# SONAR enables cell type deconvolution with spatially weighted Poisson-Gamma model for spatial transcriptomics

Zhiyuan Liu [1,2], Dafei Wu[1], Weiwei Zhai [1,2,3] & Liang Ma [1]

Recent advancements in spatial transcriptomic technologies have enabled the measurement of whole transcriptome profiles with preserved spatial context. However, limited by spatial resolution, the measured expressions at each spot are often from a mixture of multiple cells. Computational deconvolution methods designed for spatial transcriptomic data rarely make use of the valuable spatial information as well as the neighboring similarity information. Here, we propose SONAR, a Spatially weighted pOissoN-gAmma Regression model for cell-type deconvolution with spatial transcriptomic data. SONAR directly models the raw counts of spatial transcriptomic data and applies a geographically weighted regression framework that incorporates neighboring information to enhance local estimation of regional cell type composition. In addition, SONAR applies an additional elastic weighting step to adaptively filter dissimilar neighbors, which effectively prevents the introduction of local estimation bias in transition regions with sharp boundaries. We demonstrate the performance of SONAR over other state-of-the-art methods on synthetic data with various spatial patterns. We find that SONAR can accurately map region-specific cell types in real spatial transcriptomic data including mouse brain, human heart and human pancreatic ductal adenocarcinoma. We further show that SONAR can reveal the detailed distributions and fine-grained co-localization of immune cells within the microenvironment at the tumor-normal tissue margin in human liver cancer.

Organisms are comprised of various types of cells and the function of their tissues is largely determined by the cellular spatial organization[1–3]. The rapid advancement of recent spatial technologies allows for more extensive interrogation of cellular distribution over relevant spatial domains and deeper understanding of tissue functions through cellular interactions within the surrounding environment[1,4]. In particular, the sequencing-based techniques such as Slide-seq[5,6], ST[7], and Visium[8], are able to investigate larger tissue areas with more genes as compared to common image-based techniques[4,9]. Despite the increasing popularity and widespread applications, these sequencing-based techniques are generally hindered by their spatial resolution, which measures RNA transcripts in spatial locations (referred to as spots) with radius mainly ranging from 10–100 μm according to differed technologies[1]. Thus, each spot normally contains multiple cells, resulting in genes being quantified from cell mixtures. Estimate cell type composition through computational deconvolution becomes an essential step before downstream analysis to explore spatial distribution, spatial co-localization, or spatial interaction of cell types[9–12].

A number of computational algorithms, which designed specifically for deconvolution spatial transcriptomics with single-cell RNA

[1]Key Laboratory of Zoological Systematics and Evolution, Institute of Zoology, Chinese Academy of Sciences, 100101 Beijing, China. [2]University of the Chinese Academy of Sciences, 100049 Beijing, China. [3]Center for Excellence in Animal Evolution and Genetics, Chinese Academy of Sciences, 650223 Kunming, China. ✉e-mail: weiweizhai@ioz.ac.cn; maliang@ioz.ac.cn

sequencing data as references, has been recently proposed[13,14]. Methods such as SpatialDWLS[15], SPOTlight[16], and CARD[17] are based on non-negative matrix factorization (NMF) or non-negative least squares (NNLS) algorithms. These principles make underlying normality assumption and have been commonly applied to bulk tissue deconvolution (MuSiC[18], SCDC[19], DecOT[20]). Generally, Spatial transcriptomic data are often in form of counts[1,21]. Probabilistic-based methods such as RCTD[22], Cell2location[23], Stereoscope[24], model the raw counts of spatial transcriptomics with Poisson or Negative-Binomial distributions under likelihood or Bayesian framework. These probabilistic-based methods are able to incorporate statistical parameterization of various effects, such as location-specific shift and/or overdispersion of the observed expression counts. They have been demonstrated to yield robust performance in recent studies[25,26].

In spatial transcriptomic data (ST data), the resolved relative spatial localization of spots provide valuable information about neighboring context[17,27]. The Tobler's first law of geography, 'Everything is related to everything else, but near things are more related than distant things', also applies to biological systems[28,29]. For example, similar cell type compositions are likely to be identified within proximal tissue domains, where tissues are organized communities of cells that work together to perform specific functions[2,30]. Study of *Drosophila* embryo also revealed that physically closed cells tend to have similar expression profiles and vice versa[31]. The spatial information provided in ST data have been utilized to aid spatial clustering or segmentation algorithms, such as STAGATE[32], BayesSpace[33], and FICT[34], with enhanced performance. However, such precious information is rarely applied to deconvolution methods designed for ST data.

To the best of our knowledge, CARD[17] and SD2[35] are the few deconvolution algorithms that make explicit use of spatial information. CARD models spatial dependencies of cell type proportions by a conditional autoregressive model with a Gaussian kernel, where the spatial covariance structure is fixed and depends only on pairwise distance. SD2 applies a graph convolutional network (GCN) framework and utilizes spatial information by connecting each spot to its four nearest spatial neighbors in an unweighted graph. Note that the spatial information should be used with caution among tissues with heterogeneous pattern. The cellular spatial organization of tissues can be drastically altered by disease and/or infectious and inflammatory processes[9]. Specifically, there exists the so-called edge effect, which refers to a greater diversity at the boundary of communities, between tissue regions with diverse functions[36]. For instance, the cell type composition varies dramatically in different layers of mouse cortex[37] and in the tumor microenvironment at the tumor-normal margin[36]. The spatial relatedness will vanish around these boundaries due to strong edge effect. In such cases, assigning fixed weights to neighboring spots, as CARD does, or aggregating spots uniformly in the neighborhood as SD2 does, can lead to biased estimates of cell type composition in regions near the boundary between heterogeneous patches. The bias can be further propagated to local areas through conditional autoregressive modeling or deep GCN framework.

In this study, to fill in the aforementioned gaps, we propose a Spatially weighted pOissoN-gAmma Regression model, termed SONAR, for cell-type deconvolution with spatial transcriptomic data. SONAR has three main properties. First, SONAR directly models the raw counts in ST data directly with a Poisson-Gamma mixture distribution, which is able to account for the overdispersion effect. Second, in order to make full use of the rich spatial information, SONAR applies a geographically weighted regression framework, which estimates cell type composition at each spatial spot by borrowing information from neighbors. Third, to avoid bias due to the edge effects, SONAR includes a pre-clustering step and incorporates an elastic weighting strategy to adaptively tune the spatial kernel weights according to expression similarity. These steps will effectively filter out

dissimilar neighboring spots that are unlikely to share common cell type composition. By comprehensive simulation studies, we demonstrate the performance of SONAR over other state-of-the-art deconvolution methods. Furthermore, we show that SONAR can accurately map regional specific cell types in real ST data, including mouse brain, human heart as well as human pancreatic ductal adenocarcinoma. Finally, we show that SONAR can reveal the detailed distributions and co-localization dynamics of immune cells around tumor leading edge regions in liver cancer tissues.

## Results
### Overview of SONAR
In this section, we give a brief introduction to SONAR. A detailed description of the algorithm can be found in the "Methods" section.

SONAR takes the expression count matrix $Y = (y_{i,g})_{I \times G}$ of $I$ spots with $G$ genes as input (Fig. 1a) and models each $y_{i,g}$ by a Poisson-Gamma distribution (Fig. 1b). The Poisson mean is determined by the product of three terms: the total depth $\Phi_i$ of spot $i$, the expected rate of expression $\lambda_{i,g}$, and a random variable $V_{i,g}$, which accounts for the over-dispersion effect[21] and is characterized by a Gamma distribution with both parameters equal to $\alpha_i$ (Fig. 1b). Specifically, by given the reference cell type signature matrix of $T$ cell types, $S = (s_{t,g})_{T \times G}$, the expected rate $\lambda_{i,g}$ is modeled as a linear mixture of $s_{t,g}$ for all cell types with parameters $\boldsymbol{\beta}_i = \{\beta_{i,j} : j = 0, 1, 2, \cdots, T\}$. Under this model the spot-specific parameters $\boldsymbol{\beta}_i$ and $\alpha_i$ can be estimated by log-likelihood $L_i(\boldsymbol{\beta}_i, \alpha_i)$ (see "Methods" section for details).

Since spatially adjacent regions are prone to have similar compositions of cell types[30,32,33]. SONAR employs the geographically weighted likelihood principle which allows the spot-specific parameters to be calibrated by borrowing information from neighboring spots[38,39]. The parameters at each spot $i$ are thus estimated by maximizing the weighted local likelihood function $T(\boldsymbol{\beta}_i, \alpha_i)$ over spots in the local set $\mathcal{N}_i$, which constitutes of the focal spot $i$ and its neighboring spots under consideration (central equation in Fig. 1, "Methods" section). The spatial weights are modeled by SONAR with a bi-squared kernel function and are additionally tuned according to expression similarity (Fig. 1c, see "Methods" section for details). Specifically, by performing a pre-clustering step, SONAR will give zero weights to neighboring spots that have differed cluster assignment with the focal spot. The non-zero weights will be further adjusted, by elastic weighting, according to pairwise similarities of spots (Fig. 1c). These adjustments will effectively prevent additional bias caused by edge effects in tissues with complex spatial patterns[25,36,37].

Finally, SONAR outputs the spatial map of cell types and gives the detailed composition of each spot (Fig. 1d). In the following sections, we will make an in-depth evaluation of SONAR on synthetic and real spatial transcriptomic data, and benchmark it to state-of-the-art methods including RCTD, CARD, Cell2location, Stereoscope, SPOTlight, SpatialDWLS, and SD2.

### SONAR performed robustly on synthetic data
In order to systematically evaluate the performance of SONAR, we first performed simulation studies on a set of pseudo spatial data generated based on single-cell RNA-seq data ("Methods" section). We divided our simulation into two schemes according to spatial homogeneity patterns. In the first scheme, termed as Homo-Area, we aimed to test local factors under a continuous region with homogeneous (similar) cell type composition across spots. In the second scheme, named as Compo-Area, we intended to imitate more general spatial structures in composite regions, which consist of multiple homogeneous subregions with various organization patterns and transition modes.

Under Homo-Area scheme, we designed a 20 by 20 (spots) region with cells sampled at each spot from the same distribution of

cell type compositions. We generated 80 datasets to assess the impact of local factors including the abundance of the dominant cells, the number of dominant types, the relative proportion of multiple dominant types and the abundance of sparse cells (Fig. 2a and Supplementary Fig. 1). SONAR was consistently a robust performer under various density and cell type proportion settings, while RCTD, CARD, and SpatialDWLS also showed comparably good performance (Fig. 2b and Supplementary Fig. 2). These methods were at the top of the overall mean rank scores, with SONAR outperforming others (Fig. 2c).

Homo-Area scheme demonstrated SONAR's advantages in dealing with a single homogeneous region. Nevertheless, in reality, spatial patterns of tissues are more complicated and, in general, are formed by multiple subregions[36]. The arrangement regions can be in the form of Layer pattern[13,40], Block pattern[32,34], or Background pattern, where one or more small patches inter-dispersed on a large background of homogeneous region[40,41] ("Methods" section, Supplementary Fig. 3a). In addition to spatial patterns, the transition modes between subregions also differ[36]. There can be a hard boundary between adjacent regions, which we termed as Jump-transition. There are also soft boundaries with gradient buffers[36] or mixed buffers[37], we respectively named them as Gradient-transition and Mix-transition ("Methods" section, Supplementary Fig. 3b).

We extended the region size to 20 by 40 spots in the Compo-Area scheme. We generated 95 datasets with varying perspectives of transition modes, spatial patterns, and cell abundance between regions ("Methods" section). We included in the benchmark a naïve version of SONAR, denoted as SONAR-0, which performed without pre-clustering

and elastic weighting. Most algorithms are relatively robust under the Compo-Area scheme (Supplementary Fig. 4). The overall performance of SONAR (both versions), CARD, RCTD, and Cell2location are consistently good, with SONAR consistently having the lowest estimated error and ranking first in terms of overall mean rank (Fig. 2d and Supplementary Fig. 4).

Interestingly, we observed apparent performance differences in the two versions of SONAR under different transition modes due to the edge effect, with the full version of SONAR being significantly better (Fig. 2e). When there exist Jump-transition between subregions, we found that SONAR-0 produces low and smoothed estimation error within homogeneous regions, but sharp errors at the border of two subregions (Fig. 2f). CARD, which models spatial dependency according to only spatial distance, also tends to spill errors into neighboring areas. For instance, in the orange subregions in layer pattern (second layer from the bottom) and in block pattern (top left block) examples (Fig. 2f), dispersed errors were found in CARD but not in SONAR, RCTD or Cell2location (Supplementary Figs. 5a and 6a). The elastic weighting design in SONAR made the involvement of local neighbors more flexible. The exclusion of spots less similar to the focal spot significantly mitigated the edge effect displayed in the results of SONAR-0. The addition of a pre-clustering step in SONAR further improves the overall JSD (Fig. 2f, g and Supplementary Figs. 5b and 6b).

In summary, our simulation study demonstrated the ability of SONAR in deconvolution under complex spatial patterns and transition modes.

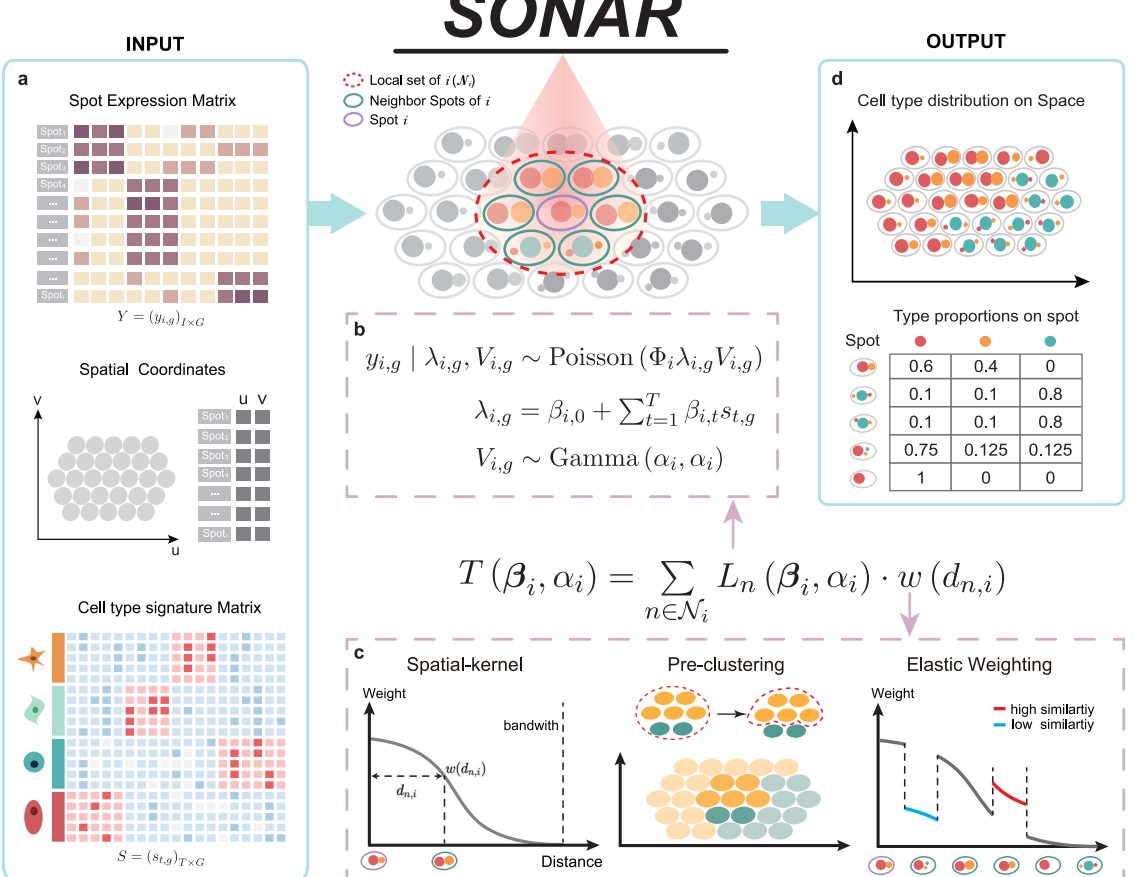

**Fig. 1 | Overview of the computational framework of SONAR.** SONAR is designed for cell-type deconvolution of spatial transcriptomic data. **a** SONAR inputs spatial transcriptomics expression matrix (top) with coordinates information (middle) and an annotated scRNA-seq data as cell type reference (bottom). **b** The Poisson-Gamma regression model of SONAR. **c** The spatial (kernel) weight and the adaptive tuning strategy (pre-clustering, and elastic weighting) applied in SONAR. **d** SONAR outputs a spatial map of cell types and the detailed cell type composition of each spot.

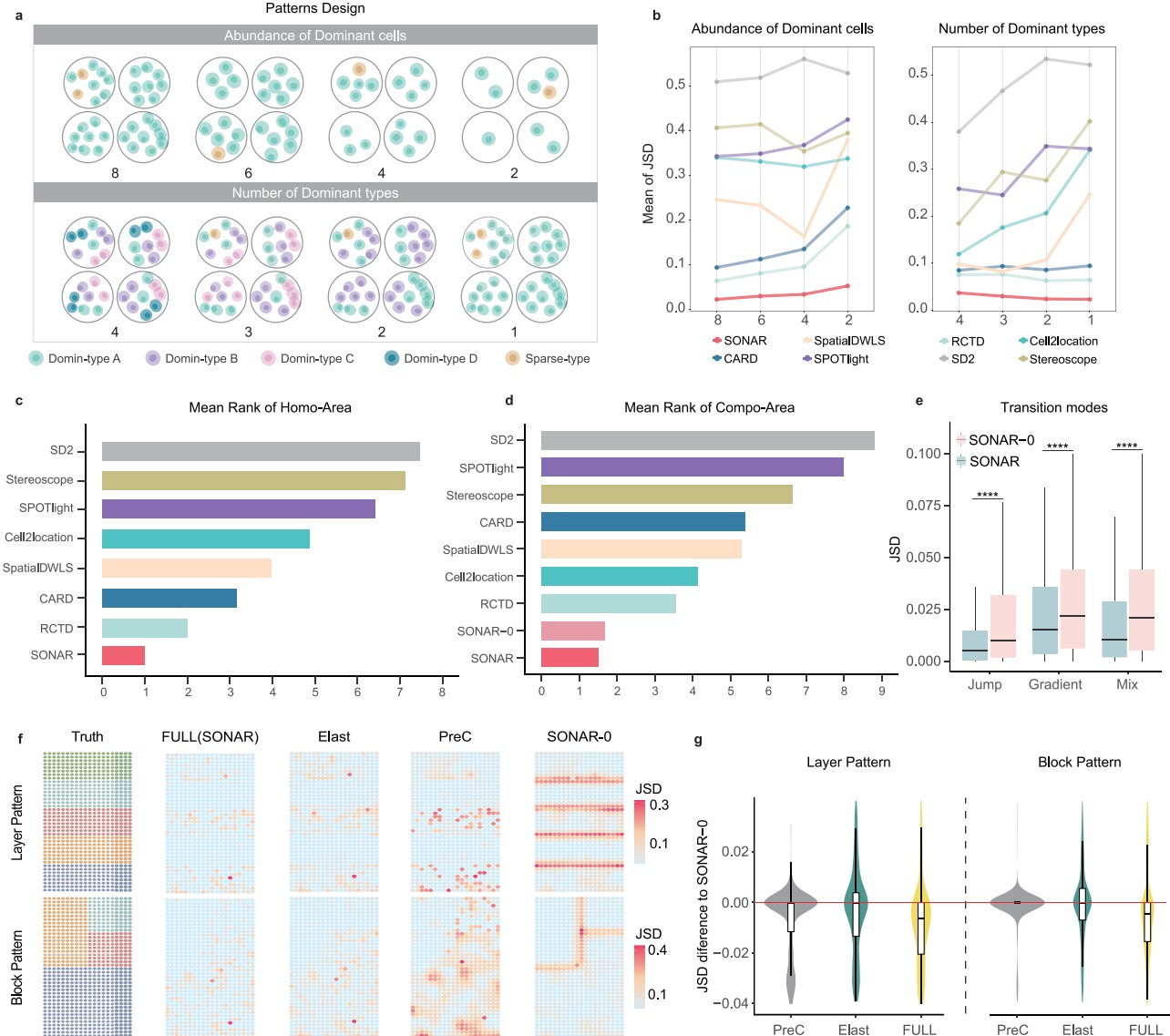

**Fig. 2 | Benchmarking SONAR under synthetic data. a** Pattern design under Homo-Area scheme. The first row, the abundance of dominant cells decreases from 8 to 2. The second row, the number of dominant cell types changes from 4 to 1. **b** The mean JSD of all compared methods under synthetic data simulated according to scenarios in **a** ($n = 2000$, 5 replicates * 400 spots). **c**, **d** The sorted MR scores (which is aggregated over RMSE and JSD across all scenarios; see Methods) of Homo-Area scheme and Compo-Area scheme. **e** The boxplots comparing JSD of SONAR to SONAR-0 (the raw version of SONAR without performing pre-clustering and elastic weighting) under the three different transition mode ($n = 4000$, 5 replicates * 800 spots). Tested under paired one-sided (less) Wilcoxon rank-sum

test. ****$p \leq 0.0001$. The exact $p$-values are $7.4 \times 10^{-47}$, $7.7 \times 10^{-6}$, $1.6 \times 10^{-6}$ respectively. **f**, **g** Exemplar comparison among different versions of SONAR under Layer and Block patterns. PreC is SONAR-0 with pre-clustering, Elast is SONAR-0 with elastic weighting, and FULL is SONAR. The first column show the true pattern and other columns show the JSD at each spatial spot. In **g**, the JSD difference of each spot for the exemplar scenarios in **f**, with SONAR-0 as the baseline (the red horizontal line represents no differences, $n = 800$ spots). Each box plot in **e** and **g** ranges from the first and third quartiles with the median as the horizontal line, while whiskers represent 1.5 times the interquartile range from the lower and upper bounds of the box. Source data are provided as a Source Data file.

## SONAR maps layer-specific neurons in mouse cortex and locates fine scale structure in human heart

To assess SONAR under realistic tissues with complex patterns, we evaluated its performance on real ST data with known annotation or with clear regional structure.

First we applied SONAR to a Mouse visual cortex data which is in the form of a typical layer pattern. The data was acquired by STAR-map, which achieves single-cell resolution[42]. It includes 1549 cells that correspond to 15 cell types, and was gridded into 189 pseudo-spots with each containing 1–18 cells[26] ("Methods" section). We applied scRNA-seq dataset from ref. 43 as reference. SONAR could clearly distinguish cell types in different layers (Fig. 3a and Supplementary Fig. 7). The Glutamatergic neurons,

such as L2/3, L4, L5, and L6 neurons, dominate in spatially distinct but partially overlapping layers[37]. SONAR successfully recovered these regionally densely distributed neuron types with high correlations of estimated to true cell type proportion (Fig. 3a and Supplementary Fig. 7 and Table 1). The GABAergic neurons (Pvalb, Sst, and Vip) are more loosely distributed and are often located within one or more layers[37]. SONAR also resolved these sparse neuron types and many other Non-neuron types, such as Olig and Smc cells (Fig. 3a and Supplementary Fig. 7 and Table 1). Compared to other methods, SONAR had the lowest MR score; SONAR-0 and RCTD tied for the second place (Fig. 3b). SONAR also outperformed other methods in overall correlations of estimated cell proportions to the truth (Table 1).

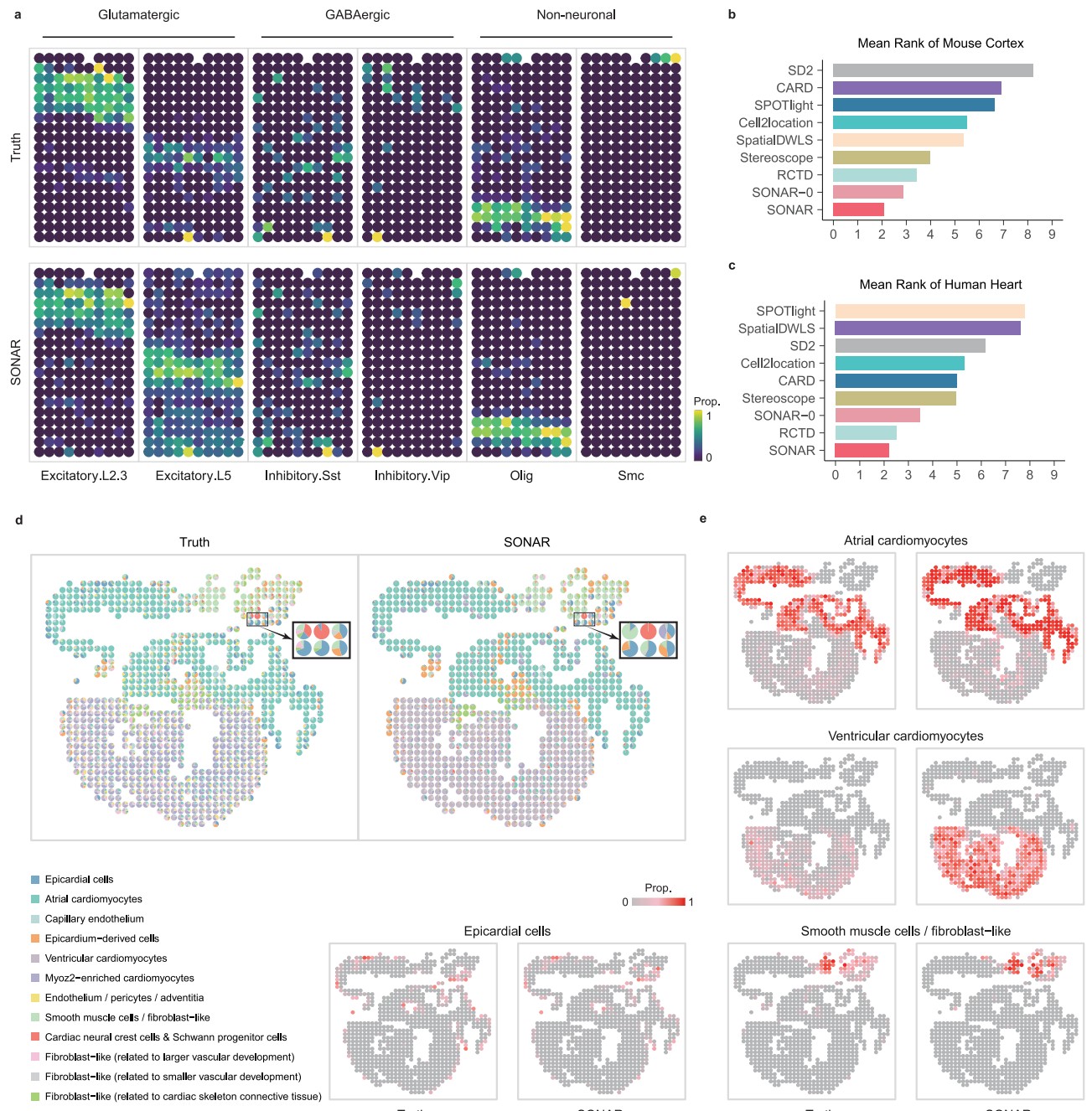

**Fig. 3 | Evaluation of real spatial transcriptomics data. a** Truth (top row) vs SONAR predicted (second row) proportions of Glutamatergic neurons (left 2), GABAergic neurons (middle 2), and Non-neuron cell types (right 2) in the mouse visual cortex dataset. **b**, **c** The sorted MR scores of compared algorithms in mouse visual cortex dataset and human heart dataset. **d** Pie plots of cell type composition at each spatial spot in human heart dataset, ground truth vs SONAR predictions. The magnified boxes show the unique distribution of Cardiac neural crest cells (cNCCs) and Schwann progenitor cells (SPCs) in the outflow tract region. **e** The comparison between the ground truth and inferred proportions by SONAR of the 4 major cardiac cell types (atrial cardiomyocytes, ventricular cardiomyocytes, smooth muscle cells/fibroblast-like, and epicardial cells). The colors represent the scaled proportion for each cell type. Source data are provided as a Source Data file.

We further performed SONAR to mouse hippocampus dataset acquired by Slide-seqV2[6], which contains 41,795 spots and 5093 genes. The mouse hippocampus has two major regions: Cornu Ammonis (CA) and Dentate Gyrus (DG). The principle cell type located by SONAR were in accordance to the regions annotated in Allen Brain Institute[44]. For example, Dentate cell type was mainly located in the V-shaped granule layer in DG (Supplementary Fig. 8a). CA1 and CA3 cell types were correctly mapped to the respect pyramidal layers of CA. The pyramidal layers occupies a thin linear region, which is in the form of a typical background pattern (see "Methods" section). Other methods

such as RCTD, Cell2location, and Stereoscope were also able to map these major cell types. When evaluating the correlation between inferred cell type proportions and cell type specific markers (Supplementary Table 1), SONAR showed superior performance overall other methods in terms of overall correlation ranks (Supplementary Fig. 8b and Supplementary Table 2).

Next, we applied SONAR to Developing human heart data, which generated based on in situ sequencing[45]. It contains 17,444 cells with 65 genes, and was gridded into 1039 pseudo-spots[10] ("Methods" section). The annotated single-cell data of the human heart data were used as

**Table 1 | Correlation of estimated to true cell type proportions in Mouse cortex data**

|                  | SONAR  | SONAR-O | CARD   | RCTD   | Cell2location | Stereoscope | SPOTlight | SpatialDWLS | SD2    |
|------------------|--------|---------|--------|--------|---------------|-------------|-----------|-------------|--------|
| Olig             | 0.957  | **0.960** | **0.960** | 0.956 | 0.953       | 0.955       | 0.912     | 0.947       | 0.877  |
| Excitatory L2/3  | **0.885** | **0.884** | 0.549 | 0.878 | 0.766       | 0.678       | 0.850     | 0.400       | 0.532  |
| Inhibitory Sst   | **0.851** | 0.828  | 0.756  | 0.819  | 0.489         | **0.839**   | 0.503     | 0.712       | 0.262  |
| Excitatory L4    | 0.826  | 0.840   | 0.646  | **0.865** | 0.828      | **0.872**   | 0.795     | 0.723       | 0.349  |
| Excitatory L6    | 0.824  | **0.840** | 0.785 | 0.810  | **0.838**     | 0.723       | 0.836     | 0.699       | 0.458  |
| Astro            | **0.789** | **0.799** | 0.592 | 0.778 | 0.743       | 0.772       | 0.748     | 0.682       | 0.425  |
| Excitatory L5    | **0.711** | **0.726** | 0.515 | 0.706 | 0.654       | 0.561       | 0.552     | 0.453       | 0.172  |
| Inhibitory Pvalb | 0.627  | 0.670   | 0.659  | 0.696  | **0.709**     | **0.779**   | 0.627     | 0.585       | 0.371  |
| Endo             | **0.586** | 0.584  | 0.584  | **0.599** | 0.527      | 0.557       | 0.424     | 0.384       | 0.174  |
| Inhibitory Vip   | **0.572** | **0.566** | 0.332 | 0.310  | 0.400       | 0.426       | 0.431     | 0.294       | 0.324  |
| Smc              | **0.515** | 0.130  | **0.599** | 0.270 | 0.138       | −0.144      | −0.144    | −0.112      | −0.044 |
| Micro            | −0.060 | −0.053  | −0.132 | −0.062 | **−0.031**    | −0.034      | −0.050    | −0.069      | **0.052** |
| **Average**      | **0.673** | **0.648** | 0.570 | 0.635 | 0.585       | 0.582       | 0.540     | 0.493       | 0.329  |

The value indicates the Pearson correlation of estimated to true cell type proportions for each cell type for all algorithms. Higher values indicate better performance. The top two correlations for each cell type are shown in bold.

reference[10,45]. SONAR ranked at the top and performed better than RCTD and Stereoscope (Fig. 3c), all of which were benchmarked to have captured the expected spatial distribution of cell types[10] (Fig. 3d and Supplementary Fig. 9). SONAR-O ranked, between RCTD and Stereoscope, in the third place. SONAR correctly mapped the major cardiomyocytes the corresponding atrial and ventricular bodies (Fig. 3e). It also finely predicted the distribution of Smooth muscle cells in the outflow tract and mapped Epicardial cells to the thin outer layer of the heart (Fig. 3e).

It is worth noting that both versions of SONAR captured the key distributions of Cardiac neural crest cells (cNCCs) and Schwann progenitor cells (SPCs) which are uniquely present in the outflow tract region (Fig. 3d)[45]. This tiny structure of cell distribution, which was reported in ref. 45, was missed or misplaced by other algorithms (Supplementary Fig. 9). In addition, we further investigated the impact of different resolution and spot scale to the performance of deconvolution. We mimic pseudo-spots with higher resolution by reducing grid size to 75% and 50% of the original grid size (see Supplementary Note 2). SONAR maintained its advantage over the other methods compared, although there is a decrease in performance from the original grid size to the higher resolution size (Supplementary Fig. 10a). In particular, the distribution of cNCCs and SPCs were robustly mapped by SONAR at different resolutions. (Supplementary Note 2 and Supplementary Fig. 10b, c). The results demonstrated the ability of SONAR to capture the fine spatial structure of cell composition.

## SONAR characterized regional cell type distributions in pancreatic ductal adenocarcinoma

Next, We applied SONAR to a human pancreatic ductal adenocarcinoma (PDAC) spatial transcriptomics dataset (PDAC-A-1 in ref. 46), which were acquired from the ST technology[7]. The PDAC tissue has four major regions including the Cancer region, Ductal Epithelium region, Stroma region, and Pancreatic region (Fig. 4a), which were annotated according to features of H&E staining by histologists[17,46]. For cell type reference, we applied annotated scRNA-seq data generated from paired sample[46].

SONAR well characterized the spatial heterogeneity of cell type composition in PDAC (Fig. 4a and Supplementary Fig. 11). The cell type proportions estimated by SONAR were all significantly correlated with the expression of cell type specific marker genes (Supplementary Tables 3–5). For example, spots enriched with *TM4SF1*, *PRSS1/2*, and *SERPINA1* are also predicted to, respectively, have high proportions of Cancer clone A cells, Acinar cells, and Ductal centroacinar cells (Fig. 4b). By comparing results to other methods, SONAR showed

stronger correlation in most cell types (Fig. 4c and Supplementary Table 4). Even for widely distributed rare cell types such as DC cells or T and NK cells, SONAR can still result in significant cell type proportion to marker correlations (Fig. 4c and Supplementary Fig. 11).

SONAR correctly identified regionally restricted cell types (Fig. 4d and Supplementary Figs. 11–13). For example, in Cancer region, SONAR not only mapped the two Cancer clone cells, but also captured the enrichment of Ductal high hypoxic and Fibroblast cells (Fig. 4d top). In Pancreatic region, SONAR revealed significantly higher proportion of Acinar cells and Endocrine cells (Fig. 4d middle). Especially, the distribution of Endocrine cells in the central and upper Pancreatic subregions (Supplementary Figs. 11 and 13). In addition, SONAR identified the differential distribution of the two Dendritic cell subpopulations (Fig. 4d middle and bottom), where subpopulation A (mDCs A) was enriched in the Pancreatic region and subpopulation B (mDCs B) was enriched in the Ductal region[46]. These regional distributions that have been fully characterized by SONAR were only partially identified by other methods (Supplementary Figs. 11–13). The regional correlation of cell type proportion and expression of type-specific markers also demonstrated the superior performance of SONAR over other methods within each subregion (Supplementary Fig. 14 and Supplementary Table 5).

Better estimation of regional cell type proportions allowed SONAR to make improved characterization of regional co-localization of cell types. SONAR identified Ductal high hypoxic cells co-localize with mDCs A cells in the Cancer region (Fig. 4e and Supplementary Fig. 15a), which indicates that specific subtypes of mDCs cells may be generated by precursors exposed to hypoxia environments[47]. The co-localization of Endothelial cells and Fibroblasts revealed by SONAR in the Cancer region reflects the complex interplay of the main components in tumor stroma during tumor growth process[48] (Fig. 4e and Supplementary Fig. 15a). Moreover, SONAR also discovered the co-localization of Acinar cells with both Endocrine and mDCs A cells in the Pancreatic region (Fig. 4e and Supplementary Fig. 15b), which trends were not detected by other methods (Fig. 4e).

## SONAR can fine-mapping co-localized cell types at tumor leading edge

We further applied SONAR to resolve heterogeneity of tumor leading edge region in primary liver cancer. The four samples HCC-1L, HCC-2L, HCC-3L, and HCC-4L, which generated by 10x Visium platform[25], all constitute of Normal, Tumor, and Transition regions in one slice. We applied single-cell dataset in ref. 49, which contained ~74,000 cells and was annotated to 17 cell types, as reference.

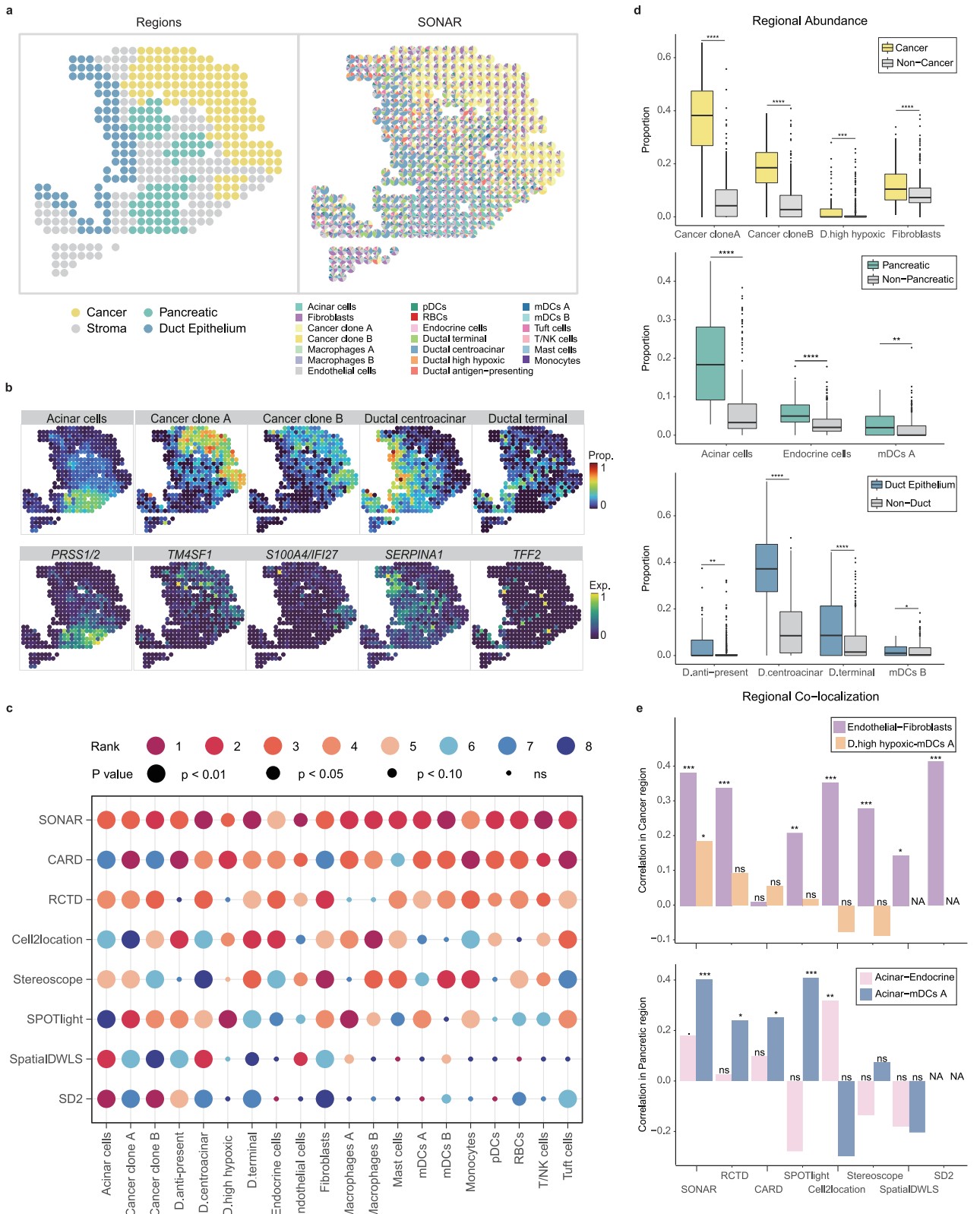

SONAR successfully resolved the spatial distribution of cell types in the samples (Supplementary Fig. 16). The proportions of Fibroblasts and Endothelial cells were mainly mapped by SONAR to the Transition region, which forms the fibrous capsule between Normal and Tumor regions[25,50,51] (Fig. 5a and Supplementary Figs. 17 and 18). The proportions of these two cell types in Normal and Tumor regions were observed to be higher in HCC-2L, which has disrupted capsule structure, as compared to the other three samples (Fig. 5b). We also discovered a significantly lower proportion of Myeloid cells in the Normal region compared to the Tumor region only in sample HCC-2L (Supplementary Fig. 17), while in HCC-1L, 3L and 4L this tendency reversed to higher. These regional distinctions between and within samples were in accordance with ref. 25, and were fully characterized only by SONAR and RCTD (Supplementary Figs. 19–24. Note that we

**Fig. 4 | Analyzing PDAC data. a** Left, annotated spatial regions of PDAC data. Right, pie plots of the inferred cell-type composition at each spot by SONAR. **b** Top, the (scaled) proportions of regional specific cell types inferred by SONAR. Bottom, the (scaled) expression levels of corresponding cell-type-specific marker genes. **c** Correlations between inferred cell-type proportions and corresponding cell-type-specific marker genes across spatial locations for each algorithm. The colors represent rank of correlations for each cell type over all algorithms. The $p$-values indicate the significance level under one-sided Spearman rank test for positive correlation. **d** Comparisons of regional abundance of regional specific cell types in three regions: Cancer region ($n = 137$ spots) vs non-Cancer region ($n = 289$ spots) (top), Pancreatic region ($n = 70$ spots) vs non-Pancreatic region ($n = 356$ spots) (middle), and Duct Epithelium region ($n = 72$ spots) vs non-Duct Epithelium region ($n = 354$ spots) (bottom), with the $p$-value is indicated by one-sided (greater) Wilcoxon rank-sum test. Each box plot ranges from the first and third quartiles with the median as the horizontal line, while whiskers represent 1.5 times the interquartile range from the lower and upper bounds of the box. **e**, The regional (Pearson's) correlation between cell-type proportions for co-localized pairs of cell types in Cancer region (top) and Pancreatic region (bottom), $p$-values indicate the significance level under one-sided $t$-test for positive correlation. ****$p \le 0.0001$, ***$p \le 0.001$, **$p \le 0.01$, *$p \le 0.05$, ·$p \le 0.1$, ns: $p > 0.1$. mDCs myeloid dendritic cells, pDCs plasmacytoid dendritic cells, RBCs red blood cells, NK cells natural killer cells, D.high hypoxic, Ductal high hypoxic; D.anti-present, Ductal antigen-presenting; D.centroacinar, Ductal centroacinar; D.terminal, Ductal terminal; Non-Duct, Non-Duct Epithelium. Source data are provided as a Source Data file including the exact $p$-values in **c**–**e**.

excluded SD2 in this comparison since its training process did not converge in all four datasets).

In addition, the cell type compositions resolved by SONAR reflected the clinical status of patients. For example, HCC-2L is in advanced tumor stage as compared to other samples[25], and was identified to have comparatively higher proportions of Tregs and B Cells (Supplementary Fig. 25). These cell types are reported to be positively correlated with progressive tumor stage[52–54], and further the Treg proportion also negatively correlates with encapsulation[55]. Moreover, the HBV-positive samples HCC-1L, HCC-2L, and HCC-4L[25] were identified to have significantly elevated B Cells proportions in Transition regions (Supplementary Fig. 17), which confirmed the observations in ref. 52 that an increased percentage of B Cells in the tumor margin region are found in HBV patients.

Interestingly, although Fibroblasts and B Cells are enriched in the Transition region of these HBV-positive samples, their regional co-localization was absent (Supplementary Figs. 17 and 26). This trend was revealed only by SONAR in all three samples (Fig. 5c and Supplementary Fig. 26). To further investigate the fine-grained co-localization of cell types, we defined a LoCo score for each spot, which characterizes the local co-localization of a pair of cell types ("Methods" section). We observed that higher Fibroblasts-B Cells LoCo scores seem to be distributed along the leading edges between Transition region and the other two regions (Fig. 5d, e). Specifically, the LoCo scores at the Outer Edge of the Transition region tend to be greater than that of the Inner Edge (Fig. 5e). Further, the LoCo scores in other regions are all significantly higher than the Interior Transition region (Fig. 5e). In addition, samples with complete capsules (HCC-1L and HCC-4L), stronger levels of co-localization were distributed at the Normal side of the Edge. While for HCC-2L, higher LoCo scores were distributed around the Tumor side of the Edge, probably due to its disrupted capsule at the Normal side (Fig. 5d, e). These fine-scaled co-localization tendencies of Fibroblasts and B Cells were more clearly revealed by SONAR as compared to other methods (Supplementary Fig. 27). In particular, mapping the regional median LoCo scores (Supplementary Figs. 28–30 and Supplementary Table 6) onto the spatial spots, allowed us to explicitly visualize the sharp Outer Edge resolved by SONAR that forms a local peak over neighboring regions. SONAR also identified similar co-localization trends between Fibroblasts and other immune cell types (Supplementary Fig. 31). These results again demonstrated the advantage of SONAR in fine-mapping of cell types at sharp boundary region in complex tissues.

## Discussion

Sequencing-based spatial transcriptomics offers an unprecedented opportunity to interrogate cellular organization and functional distribution in large tissue regions. With the accumulation of well-annotated scRNA-seq data, cell type deconvolution, and spatial mapping is becoming a critical step prior to most downstream analyses of ST data. In this study, we proposed SONAR, a spatially weighted regression with direct modeling of raw counts in ST data by Poisson-Gamma mixture distribution, which can sufficiently account for the over-dispersion effect. SONAR utilized the wealth of spatial information to resolve cell types' compositional distribution over tissue space. By synthetic data under various designs of cellular compositions and spatial patterns, we demonstrated SONAR's ability to accurately map the spatial distribution of cell types. SONAR correctly localized the regional-specific cell types in ST datasets, including mouse brain, human heart, and human PDAC. SONAR also identified the spatial co-localization trend of immune cells in the human liver tumor microenvironment. In addition, SONAR demonstrated high computational efficiency (Supplementary Note 3).

There are methods that make use of spatial information and aim to perform spatial expression reconstruction based on scRNA-seq data[1]. For example, the optimal-transport-based models novoSpaRc[56] and SpaOTsc[57]. novoSpaRc reconstructs spatial gene expression from single-cell data to a predefined physical region, based on a structural correspondence hypothesis, that is cells in physical proximity share similar gene expression profiles. SpaOTsc is specifically designed for image-based spatial expression data. It aims to complete the expression profile of each cell/spot in space and infer cell-cell communication. Other deconvolution algorithms such as SPICEMIX[58], unsupervisedly represents each spot by a mixture of latent factors (metagenes) and tries to elucidate the interplay of spatial and intrinsic factors of each spot. Although these methods map cell expression profile to spatial locations or deconvolute spatial transcriptome into latent factor factors, they are not designed typically for spatial cell-type deconvolution task. As a result, we did not compare with these methods.

The spatial information is invaluable for characterizing cellular spatial constitution in relative homogeneous tissue regions. However, inducing spatial information indiscriminately will introduce extra bias due to the edge effect when the sampled tissue area covers multiple heterogeneous subregions with sharp transition boundaries. For instance, in layered mouse visual cortex data, SONAR outperformed both SONAR-0 and CARD, which only apply fixed kernel function to incorporate spatial information. SONAR improved the performance by applying pre-clustering and elastic weighting. Thereby, the spatial information can be effectively utilized over spots with identical cluster assignment to greatly mitigate bias caused by edge effect. The adaptively tuning of spatial weights according to neighboring expression similarities will further improve local cell type deconvolution in fine scale. SONAR demonstrated its effectiveness in the application to human liver cancer samples, which cover the transition region from pure normal to pure tumor regions[25]. The sharp leading edges at tumor margin regions were clearly resolved by SONAR, with the detailed co-localization dynamics of B Cells, as well as other immune cells, to Fibroblasts finely preserved.

The choice of single-cell reference can be vital in all supervised deconvolution methods. In human heart data, for instance, we applied scRNA-seq reference sampled from a different donor that sampled the ST data. Although SONAR was able to correctly map the distribution of cNCCs and SPCs to the outflow tract region in the heart, this unique composition can merely be identified by other

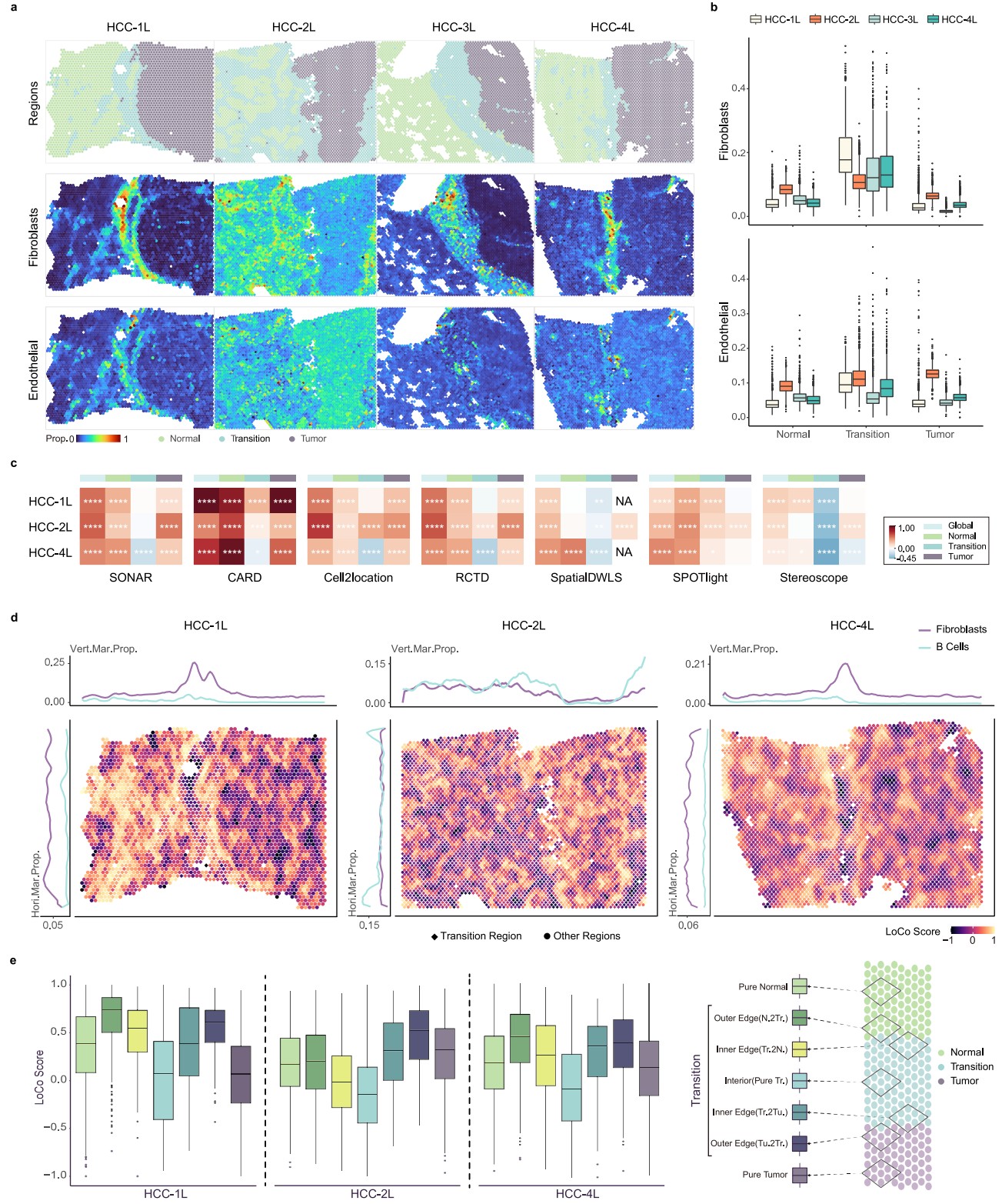

methods when using internal reference (from single cell sampled from the same donor) as shown in the recent benchmarking study[10]. Evaluation of single-cell reference is beyond the scope of our study. Nevertheless, applying reference from the paired single-cell sample, if available, is always recommended, which generally will produce better results than using external references (single-cells from the different donor)[10,20].

In summary, we proposed a model-based deconvolution algorithm SONAR, which greatly improved the accuracy in resolving cell type composition in ST data under complex spatial patterns. With the cumulation of spatial transcriptomic data, we expect that SONAR can provide fine-grained cell-type mapping of tissue architecture and aid the understanding of spatial function distributions.

## Methods

### Spatial modeling of SONAR
SONAR describes a Geographically Weighted Multivariate Poisson-Gamma Regression model for cell types deconvolution of regional

**Fig. 5 | Analyzing HCC data. a** Top, annotated regions in 4 samples (HCC-1L, HCC-2L, HCC-3L, HCC-4L): Normal, Transition, and Tumor regions. Middle and bottom, the (scaled) proportions of Fibroblasts and Endothelial cells inferred by SONAR. **b** Boxplots of regional proportions of Fibroblasts (top) and Endothelial cells (bottom) inferred by SONAR in 4 samples ($n = 1064; 1063; 1684; 1381; 409; 1922; 1083; 601; 1281; 1685; 1311; 2127$ spots from left to right boxes, respectively). **c** Correlations of Fibroblasts and B Cells proportions which inferred by each algorithms in the three HBV-positive samples (HCC-1L, HCC-2L, HCC-4L): Global, Normal, Transition, and Tumor regions. Color indicates two-sided Spearman rank correlations with $p$-values represent the significance levels: ****$p \le 0.0001$, ***$p \le 0.001$, **$p \le 0.01$, *$p \le 0.05$. **d** The spatial plots (major sub-panel) show the spatial distribution of LoCo scores of Fibroblasts and B Cells inferred by SONAR on each spots in samples HCC-1L, HCC-2L, and HCC-4L. The shapes of spots indicate the Transition region (rhombus shape) and non-Transition regions (circle shape). The line plots in the upper/left sub-panels show the marginal distribution of the (vertical/horizontal) average proportions of Fibroblasts (purple) and B Cells (blue). Vert.Mar.Prop. vertical marginal proportion, Hori.Mar.Prop. horizontal marginal proportion. **e** The stratified distribution of LoCo scores of Fibroblasts and B Cells based proportions inferred by SONAR ($n = 781; 314; 182; 120; 78; 80; 1199; 330; 727; 790; 865; 275; 295; 1386; 877; 570; 236; 175; 108; 215; 1926$ spots from left to right boxes, respectively). Spots are stratified according to the regional composition of their neighboring set into (right sub-panel): Pure Normal, Outer Edge to Transition region (Normal or Tumor side), Inner Edge to Transition region (Normal or Tumor side), Interior Transition region, and Pure Tumor region. Each box plot in **b** and **e** ranges from the first and third quartiles with the median as the horizontal line, while whiskers represent 1.5 times the interquartile range from the lower and upper bounds of the box. Source data are provided as a Source Data file including the exact $p$-values in **c**.

spatial transcriptomics data, which properly considers the similarity information of neighboring spots.

Let $\mathcal{I} = \{1,2,\cdots,I\}$ indicate the spatial spots from the ST data, with coordinates of each $i \in \mathcal{I}$ recorded as $c_i = (u_i, v_i)$. Let $\mathcal{G} = \{1,2,\cdots,G\}$ be the set of assayed genes, and the observed spatial transcriptomic count matrix denotes as $Y_{I \times G} = (y_{i,g})_{i \in \mathcal{I}, g \in \mathcal{G}}$. Further let $S_{T \times G} = (s_{t,g})_{t \in \mathcal{T}, g \in \mathcal{G}}$, denotes the reference cell type signature matrix for $T$ cell types in $\mathcal{T} = \{1,2,\cdots,T\}$ of $G$ gene, which preprocessed from annotated single-cell data (ref. 22 and see preprocessing of single cell reference). We model the expected expression rate, $\lambda_{i,g}$, of gene $g \in \mathcal{G}$ at spot $i \in \mathcal{I}$ as a linear function of reference cell type signatures $s_{t,g}, \forall t \in \mathcal{T}$:

$$\lambda_{i,g} = \beta_{i,0} + \sum_{t=1}^{T} \beta_{i,t} \cdot s_{t,g},$$

where the intercept $\beta_{i,0}$ defines the spot-specific effect, and $\beta_{i,t}$s represent the mixture weights of reference cell types $t \in \mathcal{T}$. For convenience, we refer all the unknown parameters to be estimated in the above linear model for spot $i$ as $\boldsymbol{\beta}_i = \{\beta_{i,j} : j \in \mathcal{T} \cup \{0\}\}$.

For each spot $i \in \mathcal{I}$ and gene $g \in \mathcal{G}$, we model $y_{i,g}$ by a Poisson-Gamma distribution as:

$$y_{i,g} | \lambda_{i,g}, V_{i,g} \sim \text{Poisson}(\Phi_i \lambda_{i,g} V_{i,g}), \quad (1)$$

where $\Phi_i$ is the total number of UMIs in spot $i$, which serves as scale factor for sequencing depth at each spatial location; $V_{i,g} \sim \text{Gamma}(\alpha_i, \alpha_i)$ is the unobserved multiplicative random effect of heterogeneity, which accounts for gene- and spatial-specific over-dispersion. When $\alpha_i \to \infty$ or fix $V_{i,g} = 1$, the model in Eq. (1) reduce to Poisson with mean $\Phi_i \lambda_{i,g}$. In this study we will use the over-dispersion version by default.

Under the model in equation (1), we can write the likelihood of observing $y_{i,g}$ as $P(y_{i,g} | \boldsymbol{\beta}_i, \alpha_i)$ (See Supplementary Note 1 for details). The log-likelihood of observing expression profile of all genes in $\mathcal{G}$ at spot $i, \forall i \in \mathcal{I}$ can be then written as:

$$L_i(\boldsymbol{\beta}_i, \alpha_i) = \ln \prod_{g \in \mathcal{G}} P(y_{i,g} | \boldsymbol{\beta}_i, \alpha_i). \quad (2)$$

Since spatially neighboring spots are likely to have similar cell type compositions[17], we introduce the Geographically Weighted Poisson-Gamma model. It adopts a kernel regression methodology that uses a spatial weighting function to calibrate local parameters[39,59].

We define the local set of spot $i$ as $\mathcal{N}_i = i \cup \{n : n \text{ is a neighbor spot of } i\}$. The local log-likelihood of spot $i$ can be approximated as[39,59,60]:

$$T(\boldsymbol{\beta}_i, \alpha_i) = \sum_{n \in \mathcal{N}_i} L_n(\boldsymbol{\beta}_i, \alpha_i) \cdot w(d_{n,i}), \quad (3)$$

where $L_n(\boldsymbol{\beta}_i, \alpha_i)$ has the same form as raw log-likelihood function defined in Eq. (2), except that the parameters are shared with the focal spot $i$. The $w(d_{n,i})$ is a weighting kernel function that decaying with spatial distance. We applied the bi-square kernel function, which decays rapidly with distance and thus is more conservative under situations with high heterogeneous spatial patterns:

$$w_{n,i}(d_{n,i}) = \begin{cases} \left[1 - (d_{n,i}/b)^2\right]^2 & d_{n,i} \le b \\ 0 & d_{n,i} > b \end{cases},$$

where $d_{n,i}$ is the Euclidean distance between spatial location of spot $i$ and $n \in \mathcal{N}_i$. The hyper-parameter $b$ defines the bandwidth which determines the maximum radius of neighbors. In the original geographically weighted regression frame, the optimal bandwidth can be optimized through cross-validation or Akaike information criterion[39]. Here, to avoid over smoothing the cell type proportion estimates, we use a small bandwidth that accounts for neighbors in nearest proximal (first order neighbors) by default (Supplementary Note 4). In this way, we can then obtain the MLE for non-negative parameters $\boldsymbol{\beta}_i$ and $\alpha_i$ for each spot $i$. The algorithm implemented to find the MLE is in the Supplementary Note 5.

**Elastic weighting selection of similar neighbor spots**
To better characterize patchy spatial patterns and reduce bias caused by edge effect, we introduce two additional steps in SONAR[4,36].

First, in broad scale, we perform a quick pre-clustering of spatial transcriptomic data with Louvain algorithm in standard Seurat pipeline[61]. By assigning clusters to spots, the spatial kernel weight of spots within the neighboring set $\mathcal{N}_i$ that are in different cluster to the focal spot $i$, will be set to zero. We have also evaluated several popular clustering methods with different settings of hyper-parameters, i.e. the resolution parameter. The results show that SONAR is robust to the choice of pre-clustering algorithms (Supplementary Note 6).

Second, in fine scale, we apply a elastic weighting strategy: the original kernel weights of neighboring spots in Eq. (3) are adaptively adjusted according to their expression similarity to the focal spot. Intuitively, neighboring spots with more similar expression profiles are more likely to have analogous cell type composition. The elastic weighting design will increase/decrease the kernel weights to similar/dissimilar neighbors. In detail, for spot $n \in \mathcal{N}_i$ the elastic weighting adjustment of $w_{n,i}$ is denoted as:

$$\tilde{w}_{n,i} = w_{n,i}^{\rho \cdot (1 - \text{Sim}(\mathbf{E}_n, \mathbf{E}_i))},$$

where $\rho$ is a hyperparameter that controls the adaptive strength. $\mathbf{E}_i$ and $\mathbf{E}_n$ are the expression vectors of spot $i$ and its neighbor $n$, and their cosine similarity is defined as:

$$\text{Sim}(\mathbf{E}_n, \mathbf{E}_i) = \frac{\mathbf{E}_n \cdot \mathbf{E}_i}{||\mathbf{E}_n|| \times ||\mathbf{E}_i||}.$$

### Preprocessing of singe cell reference

From the count matrix of annotated scRNA-seq reference, we first filtered out cell types with <5 cells. We then normalized the library size for each cell and aggregated the expression profiles within each type to get the mean expression. For each cell type, we selected genes with average expression above 0.02% (per cell) and at least 0.75 log fold change compared to the average expression across all cell types. These steps generated the raw cell type signature matrix $\bar{S}_{T \times G}$.

Finally, we followed the steps in ref. 22 to correct the platform effects between the reference data and the spatial data. In detail, we generated the corrected reference cell type signature matrix $S_{T \times G}$ by normalizing raw signatures $\bar{S}_{T \times G}$:

$$s_{t,g} = \bar{s}_{t,g} e^{\hat{\gamma}_g}$$

where platform effects $\hat{\gamma}_g$ can be estimated as follows: firstly, summarizing the spatial transcriptomics data as a single pseudo-bulk measurement. Secondly, estimating each cell type's proportion of this pseudo-bulk by ordinary least squares regression based on the cell type expression profile extracted from the reference. Then use the proportions as weights to sum over each type's expression profile and regard summation as the overall spatial expression profile at the reference level. Meanwhile, the pseudo-bulk vector obtained in the first step can be regarded as the overall spatial expression profile at the spatial level. Finally, the ratio of corresponding elements in these two levels can be regarded as the gene-level platform effects $\hat{\gamma}_g$ between reference and spatial data, and the detailed proof can be seen in publication[22].

### Benchmark metrics

In order to measure the performance of deconvolution algorithms on synthetic/real datasets, we adopt three metrics: Root Mean Square Error (RMSE), Jensen-Shannon Divergence (JSD), and Mean Rank (MR). The definition are as follow:

**RMSE.** We define the RMSE of estimated cell type proportions as:

$$\text{RMSE} = \sqrt{\frac{1}{S} \sum_{s=1}^{S} (\tilde{u}_{s,t} - u_{s,t})^2},$$

where $S$ is the total number of all spots, $\tilde{u}_{s,t}$ is the predicted proportion for type $t$ in spot $s$, $u_{s,t}$ is the ground truth of cell type $t$. The lower RMSE means the better prediction.

**JSD.** Jensen-Shannon divergence is the symmetric, smooth version of Kullback-Leibler (KL) divergence, and it is used to measure the similarity between two probability distributions. We denote the predicted and actual cell type distributions as $P$ and $Q$ over cell type space $\mathcal{T}$. The KL divergence is defined as :

$$\text{KL}(P \| Q) = \sum_{t \in \mathcal{T}} P(t) \log\left(\frac{P(t)}{Q(t)}\right).$$

The JSD can thus be defined as:

$$\text{JSD}(P \| Q) = \frac{1}{2} \text{KL}(P \| M) + \frac{1}{2} \text{KL}(Q \| M),$$

where $M = \frac{1}{2}(P + Q)$. The lower JSD means the better prediction.

**Mean Rank.** In order to make a robust comparison of the algorithm performance and inspired by the accuracy score defined in[26], we define a mean rank score, termed as MR, by aggregating over RMSEs and JSDs. A smaller MR indicates better overall performance.

Specifically, for a single dataset $\mathcal{D}$: MR represents the average rank of RMSE and JSD of an algorithm across all spots in $\mathcal{D}$, denoted as $\text{MR}_{\mathcal{D}}$:

$$\text{MR}_{\mathcal{D}} = \frac{1}{2}(\text{Rank}_{\text{RMSE}} + \text{Rank}_{\text{JSD}}),$$

$$\text{Rank}_{\text{RMSE}} = \frac{1}{T} \sum_{t \in \mathcal{T}} \text{Rank}_{\text{RMSE}}^{t},$$

$$\text{Rank}_{\text{JSD}} = \frac{1}{S} \sum_{s \in \mathcal{D}} \text{Rank}_{\text{JSD}}^{s},$$

where T represents the total number of cell types, S represents the total number of spots, $\text{Rank}_{\text{RMSE}}^{t}$ and $\text{Rank}_{\text{JSD}}^{s}$ are the ranks of RMSE and JSD for the algorithm on cell type $t$ and spot $s$, respectively.

For a collection of datasets $\mathcal{C}$ : MR represents the average rank of RMSE and JSD of an algorithms across all datasets in $\mathcal{C}$, denoted as $\text{MR}_{\mathcal{C}}$:u

$$\text{MR}_{\mathcal{C}} = \frac{1}{2}(\text{Rank}_{\text{RMSE}} + \text{Rank}_{\text{JSD}}),$$

$$\text{Rank}_{\text{RMSE}} = \frac{1}{P} \sum_{p \in \mathcal{C}} \text{Rank}_{\text{RMSE}}^{p},$$

$$\text{Rank}_{\text{JSD}} = \frac{1}{P} \sum_{p \in \mathcal{C}} \text{Rank}_{\text{JSD}}^{p},$$

where P represents the total number of datasets in $\mathcal{C}$, $\text{Rank}_{\text{JSD}}^{p}$ is the rank of mean JSD over all spots in dataset $p$ for the algorithm, $\text{Rank}_{\text{RMSE}}^{p}$ is the mean rank of RMSE for the algorithm over all cell types in dataset $p$.

### Generate synthetic data

**Simulation based on scRNA-seq datasets.** We applied the peripheral blood mononuclear cell (PBMC) scRNA-seq datasets and referred to the general approach for simulation of pseudo spatial data[14,16,22,24]. We generated pseudo spatial data from an annotated peripheral blood mononuclear cell (PBMC) scRNA-seq dataset, and designed various cell type composition scenarios and spatial patterns. The expression mixture of multiple cell types at each simulated spatial spot is obtained with the following procedure: First, we simulated the observed number of cells for each cell type at the spot according to the designed composition (number) of cell types. We then randomly sampled the corresponding number of cells of each type from the scRNA-seq dataset. Finally, we summed up the gene expression counts of all sampled cells to generate the mixed expression profile at that spot (more details in Supplementary Note 7). We constrained each pseudo-spot to contains at most four cell types, including dominant types (default expectation parameter 6) and/or sparse types (default expectation 0.1). We divided our simulation into two schemes with scenarios of cell type composition and spatial patterns designed as follows:

In the first scheme, we aimed to test four local factors (including abundance of dominant cells, number of dominant types, relative proportion of multiple dominant types and abundance of sparse cells) under a 20-by-20 spots region with homogeneous (same expectations) cell type composition at each spot, referred to as Homo-Area (Supplementary Fig. 1). In 'abundance of dominant cells' scenarios, we varied the expected number of cells in dominant type from 8 to 2. In 'abundance of sparse cells', we changed the expected number of cells in sparse type from 0.3 to 0. In 'number of dominant

types' scenarios, we changed the number of dominant cell types from 4 to 1. In 'relative proportion of multiple dominant types' scenarios, we varied the expected number of cells in two dominant types from even expectations to uneven expectations. We generated 5 replicates for each scenarios of cell type composition with cell types sampled randomly.

In the second scheme, we modeled more general spatial structures (including spatial pattern, transition mode, and abundance change) in the form of composite regions, referred to as Compo-Area. Under this scheme, we extended the spatial region to 20-by-40 spots size, and generated composite region that constitute of multiple homogeneous subregions. In 'transition mode' scenarios, we designed three majors mode of transition between two subregions: Jump-transition, Gradient-transition, and Mixed-transition. Jump-transition means that there is no intermediate transition area between subregions, which form a sharp and clear boundary. Gradient-transition refers to a soft boundary that form a buffer area between subregions, where the cell type composition disperse in gradient from one side of subregion to the other side. Mix-transition also form a soft boundary between subregions, but with cell type composition in the buffer area present in form of a uniform mixture over compositions of the two subregions. In 'spatial patterns' scenarios we designed three major patterns of subregions arrangement (with Jump-transition): Layer pattern, Block pattern, and Background pattern. Layer pattern refers to dividing the whole region into 2, 3, 4, and 5 layered subregions. Block pattern means that the whole area is divided into hierarchically blocked subregions. Background pattern represents one or more small subregion floating on a large background of homogeneous region. In 'abundance change' scenarios, we generate two subregions with differed expectations of dominant cells. The homogeneous subregions were by default formed by one dominant cell type and three sparse types. We generated 5 replicates for each scenarios of cell type composition with cell types sampled randomly.

**Simulation based on real spatial datasets.** To benchmark the performance of deconvolution algorithms on more real ST data, we used two single-cell resolution ST datasets[10,26]. Follow the pipeline in refs. [10,26], we simulate the 'multi-cell spot' like ST datasets by 'gridding' the single-cell resolution spatial data into pseudo spots. Specifically, we divided the original coordinate space into grids. The expression profiles of all cells in each grid were summed up as the expression vector of each pseudo-spot, and the center point of the grid was used as the new coordinate.

The Mouse visual cortex data was originally acquired by STARmap[42], including 1549 cells that correspond to 15 cell types, and was gridded into 189 square pseudo-spots with each containing 1–18 cells. The scRNA-seq dataset based on Smart-seq from[43] was as reference.

The Developing human heart data was originally generated based on in situ sequencing (ISS)[45]. It contains 17,444 cells with 65 genes (4 genes filtered), and was grided into 1039 pseudo-spot with each of size 454*424 pixels. The annotated 10x scRNA-seq data of human heart[10,45] were used as reference. To assess the effect of spot scale, we further gridded the data with reduced sizes corresponding to 75% ($356 \times 318$ pixels, with an average of 10 cells) and 50% ($227 \times 212$ pixels, with an average of 4.9 cells) of the original grid size (see Supplementary Note 2).

**LoCo score.** We define LoCo score for each spot to characterize the local co-localization cell type pairs. The LoCo score at spot $i \in \mathcal{I}$ is defined as: the Spearman correlation of proportions of the two cell types over spots within the neighboring set $\mathcal{N}_i$.

**Implementation of alternative algorithms**
We compared SONAR with seven other algorithms, RCTD[22], CARD[17], SPOTlight[16], Stereoscope[24], Cell2location[23], SpatialDWLS[15], and SD2[35].

For all algorithms, we followed the corresponding tutorials on Github and used the recommended default parameter settings. For RCTD, we used the "full" mode for all datasets except for the Slide-seqV2 dataset, for which we used the "doublet" mode as recommended. For Stereoscope, Cell2location and SpatialDWLS, we use the run scripts from benchmarking pipeline[26]. Specifically, for Stereoscope, we set the parameter as: The single-cell model was trained with parameters "max epochs" = 50,000, and The spatial model was trained with parameters "max epochs" = 50,000. For Cell2location, we set "N cells per location" = 10. For SpatialDWLS, we use "n cell" = 10. For SD2, we set "spot num" = 300, "lower cellnum" = 10, "upper cellnum" = 20.

**Statistics and reproducibility**
In this study, 5 independent random replicates were produced for each designed scenario in simulation and were all included in the analysis. For real data applications, no statistical method was used to predetermine sample size, SONAR was evaluated across 8 spatial transcriptomic datasets (including mouse cortex, mouse hippocampus, human heart, PDAC, and 4 liver datasets), and no data were excluded from the analysis. The material for reproducing the results within Figures and Supplementary Figures is available in the Source Data files. In the comparison, all the methods were blinded to the ground truth of the spatial data. The outputs from the methods were then compared to the ground truth available in the respective datasets.

**Reporting summary**
Further information on research design is available in the Nature Portfolio Reporting Summary linked to this article.

## Data availability
This study made use of publicly available datasets. The data were acquired from the following websites or accession numbers. Annotated PBMC scRNA-seq data (materials of synthetic datasets for benchmarking) are publicly available at [https://github.com/MarcElosua/SPOTlight_deconvolution_analysis/tree/master/analysis/tool_benchmarking]. This data is original from the paper[14]. Gridded mouse visual cortex spatial transcriptomic data, annotated scRNA-seq reference, and the results of other algorithms on this dataset are provided by the platform[26] at [https://github.com/QuKunLab/SpatialBenchmarking/tree/main/FigureData/Figure4/Dataset10_STARmap]. These raw data are derived from the publications[42,43]. The gridded human heart data and annotated scRNA-seq reference are publicly available at [https://github.com/JiawenChenn/St-review/tree/main/processed_data/heart/ISS]. These raw data are sourced from ref. [45]. PDAC spatial datasets and the paired reference are available at GSE111672. The processed liver cancer spatial data from ref. [25] are publicly available at [http://lifeome.net/supp/livercancer-st/data.htm]. These raw data are derived from ref. [25]. The scRNA-seq reference in liver cancer analysis is available at Mendeley data [https://doi.org/10.17632/6wmzcskt6k.1]. The mouse hippocampus Slide-seqV2 dataset and annotated scRNA-seq data are available at [https://singlecell.broadinstitute.org/single_cell/study/SCP948/robust-decomposition-of-cell-type-mixtures-in-spatial-transcriptomics]. Source data are provided with this paper. All data supporting the findings of this study are available within the article and its supplementary files. Any additional requests for information can be directed to, and will be fulfilled by, the lead contact. Source data are provided with this paper.

## Code availability
The R package of SONAR is available at https://github.com/lzygenomics/SONAR[62].

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

## Acknowledgements

M.L. is supported by the National Key R&D Program of China (grant no. 2019YFA0709501) and National Natural Science Foundation of China (grant no. 11971459). W.Z. is supported in part by the Strategic Priority Research Program of the Chinese Academy of Sciences (grant no. XDPB17), the National Natural Science Foundation of China (grant no. 31970566), and National Key R&D program of China (grant no. 2018YFC1406902 and 2018YFC0910400).

## Author contributions

L.M. conceived the project. L.M. and W.Z. Supervised the study and provided funding support. Z.L. developed the model and implemented the model to simulation and real data. Z.L. and D.W. prepared and processed the real data. L.M., W.Z., and D.W. provided the computational resource. L.M. and Z.L. wrote the manuscript. All authors read and approved the final manuscript.

## Competing interests

The authors declare no competing interests.
