## [Peer Review File · Nature Communications]

SONAR enables cell type deconvolution with spatially weighted Poisson-Gamma model for spatial transcriptomicsREVIEWER COMMENTS

Reviewer #1 (Remarks to the Author):

In this article, the authors propose a Spatial weighted pOisson-gamma regression model, referred to as SONAR, for cell-type deconvolution with spatial transcriptomic data. Currently, many works have been developed for spatial transcriptomics deconvolution. SONAR implemented a geographically weighted regression to perform spatial deconvolution. I have the following comments.

1. SONAR implements a geographically weighted likelihood to borrow information from neighboring spots. It is well known that geographically weighted regression (GWR) tends to produce unsmooth surfaces due to the varying bandwidth problem. The authors need to justify this point about how you deal with this problem in the original GWR.
2. SONAR depends on the initialization of pre-clustering. The authors need to show the robustness of initialization using other clustering methods. The reason is that Louvain is not standard in spatial transcriptomics data analysis.
3. In the simulations, the authors only considered using count matrix in real data, with limited information about simulation details. It would be more informative and helpful generating the simulated admixture of multiple cell types. Moreover, different ST technologies have distinct features in spatial resolution. It would be helpful to see the impact of resolution and spot scale in simulations. Other metrics such as Pearson's correlation should be evaluated at least in simulations.
4. As a method paper, it would be more helpful using some widely used dataset in spatial transcriptomics, especially the ones in brain regions, which can be validated partially using information from Allen Brain Atlas. In addition, real data should include a larger dataset, e.g., the ones from slide-seq.

Other than those comments, I have some minor comments:

1. The order of subfigure a, b, c, d, e, f in figure 2 is not easy to understand. In general, it's order should be from left to right, or from top to bottom.
2. Can you explain why you use the mean rank rather than only use RMSE or JSD, compared to the RMSE or JSD, what's the advantages of Mean rank over RMSE or JSD.
3. In github, can you explain how to use your software in detail. For example give a brief example to show how to use your software like BayesSpace
<https://edward130603.github.io/BayesSpace/articles/BayesSpace.html>
4. The word "composion" should be composition in Figure 5e caption
5. In this paper, the author introduces methods SONAR with two versions (SONAR and SONAR-0). In the real data analysis, how do we choose them, is there an automatic way to choose it.

6. Some terms in the reference need some change. For example, in the reference 6, slide-seqv2 should be Slide-seqV2. In the reference 13, dna should be DNA. In the reference 14, 19, rna should be RNA. In the reference 44, rna-seq should be RNA-Seq. In the reference 52,53, t cell should be T cell. In the reference 50, b cells should be B cells, cd40/cd154 should be CD40/CD154.

Reviewer #2 (Remarks to the Author):

This study introduced a computational method called SONAR for cellular deconvolution problem of spatial transcriptomics data by using a spatially weighted Poisson-gamma regression model. SONAR utilized an elastic weighting step to adaptively filter dissimilar neighbors which could identify the boundary of different regions more accurately. SONAR has been benchmarked on different datasets and the performance was over the other state-of-the-art methods. Overall, this study showed novelty in method development and conducted comprehensive experiments on different datasets. However, there are still several issues that need to be addressed.

1. The authors claimed that CARD was the only deconvolution method that utilized spatial information of spots. But to my knowledge, there are also some methods used the spatial information including SpiceMix, SD2, SpaOTsc and novoSpaRc. The authors should also discuss about and compare with these studies.

2. In the experiments of Developing human heart data, it is great to see that SONAR could identify tiny structures like cNCCs and SPCs. I wonder if there would be the similar results when authors grid the cells into different resolutions of spots. Authors could try different resolutions of spots and use the quantified metrics to evaluate the performance which would directly show the benchmarking results through different methods.

3. Since authors simulated the pseudo-spots by randomly sampling the cells from PBMC scRNA-seq datasets, how was the Tobler's first law of geography authors mentioned embodied in the simulated dataset? The neighbored pseudo-spots should have similar expression profiles. Authors should consider this in a more systematic way and regulate the pipeline of simulation.

4. In the benchmarking experiments, authors could record the computational time to test the efficiency of SONAR and other compared methods.

5. The authors said 'We also discovered a significantly lower proportion of Myeloid cells in the Normal region compared to the Tumor region only in sample HCC-2L (Supplementary Fig. 15). These regional distinctions between and within samples were in accordance with (25), but were not fully characterized by other methods (Supplementary Fig. 16-21).' But as the figures showed, CARD, RCTD and stereoscope also had the similar results of Myeloid cells in HCC-2L. Authors should explain this inconsistency between the statements and figures.

6. In the experiments of primary liver cancer, authors could visualize the results of SONAR and other compared methods which will give the intuitive results to prove that SONAR could fine-mapping the sharp boundary regions in heterogeneous tissues.

7. Have authors tried large-scale spatial transcriptomics datasets, such as Slide-seq V2 which may contain more than tens of thousands of spots? I wonder about the performance of SONAR on large-scale datasets compared with other state-of-the-art methods.

8. Following current definition of RMSE, some cell types with high abundance may dominate the metric which means it may favor an algorithm that predicts well cell types with high proportion. Authors need to redefine the RMSE.

9. In the pipeline of pre-clustering, authors claimed that they used the Louvain algorithm. I wonder which resolution authors used. Because the choice of resolution affects the number of clusters. Is it fixed in the pipeline or regulated manually as the hyperparameter through each experiment?

We are very grateful to the reviewers for their insightful comments, which helped us to significantly improve our paper. We have revised the manuscript according to their comments and addressed their concerns with a point-by-point response in below. In the following response letter, we use black text for reviewer's comment, blue text for our responses to the comments and *red italic text for revisions in our manuscript*.

Point-by point Responses to Reviewers' Comments:

Reviewer #1 (Remarks to the Author):

In this article, the authors propose a Spatial weighted pOissoN-gAmma regression model, referred to as SONAR, for cell-type deconvolution with spatial transcriptomic data. Currently, many works have been developed for spatial transcriptomics deconvolution. SONAR implemented a geographically weighted regression to perform spatial deconvolution. I have the following comments.

Response: Thank you for you summary and constructive comments. We have addressed all comments with the point-by-point responses as follows.

1. SONAR implements a geographically weighted likelihood to borrow information from neighboring spots. It is well known that geographically weighted regression (GWR) tends to produce unsmooth surfaces due to the varying bandwidth problem. The authors need to justify this point about how you deal with this problem in the original GWR.

Response: Thank you for your comment. The original GWR framework applies a single bandwidth and implicitly assumes that each dependent variable operates at the same spatial scale (i.e. same bandwidth for each variable). This may pose varying bandwidth problem in cases when variables have differed scales, since the (optimal) single bandwidth only reflects the average multiple spatial scales. Recent developments in GWR have proposed a multiscale GWR or MGWR, which allows different bandwidths for each individual variable. However, the current MGWR framework is limited to Gaussian models and is still computationally expensive.

In light of the reviewer's comment, we examined the effect of bandwidth on individual cell types with simulation data. We find that, in most cell types, both versions of SONAR performed better with a small bandwidth, which only account for the neighbors in nearest proximal (first-order). The only cell type that tend to favor large bandwidth has the largest Moran's I value, which is as expected (**Supplementary Table 8**). However, setting a larger bandwidth tends to over-smooth the estimated parameter surface in all cell types (**Supplementary Fig. 33b**), especially those estimated by SONAR-0. Such overly smoothed estimates not only introduce greater amount of estimation bias, but also distort important local structures, such as region boundaries or spatially dispersed cell types (**Supplementary Fig. 33b, c Type 2, Fig. 2f**). The extra steps of pre-clustering and elastic weighting in SONAR mitigate the errors caused by over smoothness. Comparatively, we find the overall performance of SONAR to be robustly good when choosing a small bandwidth in both simulation (**Supplementary Fig. 33a, c**) and real data (**Supplementary Fig. 33d**).

We added more discussion on bandwidth in corresponding part in manuscripts and added **Supplementary Table 8** and **Supplementary Fig. 33**:
In **Methods** section in page 10 line 384:

“The hyper-parameter b defines the bandwidth which determines the maximum radius of neighbors. In the original geographically weighted regression frame, the optimal bandwidth can be optimized through cross-validation or Akaike information criterion. Here, to avoid over smoothing the cell type proportion estimates, we use a small bandwidth that accounts for neighbors in nearest proximal (first order neighbors) by default (Supplementary Note 4).”

Supplementary Note 4: Selection of bandwidth:

“In the kernel function, the bandwidth is the farthest neighbor that can be utilized and acts as a coefficient to normalize the distance. The larger the bandwidth, the more distant neighbors will be used, but the more distant neighbors may have less similarity with the target point, which may introduce unnecessary noise and increase the computational pressure of the algorithm. Therefore, it is necessary to choose an appropriate bandwidth.

The bandwidth determines the maximum radius of neighbors. In the original geographically weighted regression framework, the optimal bandwidth can be optimized through cross-validation or the Akaike information criterion.

We tested the effect of different bandwidth settings based on simulation data of Background and Block scenarios, which have complex patterns (Supplementary Fig. 3a bottom of Background/Block pattern). We found that even with a small bandwidth ($b=1.2$), the performance of the algorithm could be significantly improved by using neighbor information. Increasing bandwidth did not make an explicit improvement in performance but significantly increased computation time (Supplementary Fig. 33a).

Recent developments have proposed a multiscale geographically weighted regression or MGWR, which allows different bandwidths for each individual variable. However, the current MGWR framework is limited to Gaussian models and is still computationally expensive. We further examined the effect of bandwidth on individual cell types using simulation data with a block pattern (Supplementary Fig. 3a bottom of Block Pattern). We find that, in most cell types, both versions of SONAR performed better with a small bandwidth, which only accounted for the neighbors in nearest proximal (first-order). The only cell type that tended to favor large bandwidth had the largest Moran's I value, which is as expected (Supplementary Table 8). However, setting a larger bandwidth tends to over-smooth the estimated parameter surface in all cell types (Supplementary Fig. 33b), especially those estimated by SONAR-0. Such overly smoothed estimates not only introduce greater amount of estimation bias, but also distort important local structures, such as region boundaries or spatially dispersed cell types (Supplementary Fig. 33b,c Type 2 and Fig. 2f). The extra steps of pre-clustering and elastic weighting in SONAR mitigate the errors caused by over smoothness.

Finally, we tested different bandwidth settings on mouse visual cortex data. The result also favors a small bandwidth in SONAR ($b=1.2$, Supplementary Fig. 33d).

In sum, we choose to use 1.2 as the default bandwidth which accounts for the neighbors in nearest proximal (first-order neighbors).”

Supplementary Fig. 33

Effect of bandwidth on Simulation data and Mouse visual cortex data.

a, Effects of different bandwidths on accuracy and speed in 4 simulation datasets. The bandwidth with no neighbor information is denoted as 0. The distance for the nearest pair of spots is denoted as 1. The boxplot shows the JSD for each spot under the different bandwidths (0, 1.2, 2.2, 3.2, 4.2) in the block and background patterns (4 datasets which each contains 800 spots). The barplot represents the computation times of different bandwidths relative to bandwidth 0. The colors represent different bandwidths. **b**, Spatial scatter plots of the true and predicted proportions for each cell type in the Block pattern simulation under different bandwidths. The first row shows the true proportions (left) and the predictions without neighbor information (right). The second and third rows show the predictions by SONAR and SONAR-0 with bandwidths of 1.2 and 4.2, respectively. The color and size of the scatters indicate the proportions for each spot. **c**, Boxplots of the absolute errors for each cell type under different bandwidths. The colors represent SONAR and SONAR-0. The absolute error is calculated by the absolute difference between the true and predicted proportions for each spot. The rightmost column summarizes the Jensen-Shannon divergence (JSD) for all spots and all cell types. **d**, Boxplots of the JSD for the mouse brain dataset under different bandwidths, 0 to 4.2. The colors represent SONAR and SONAR-0.

Supplementary Table 8: Moran's I for All cell types in Simulation data						
	Cell Type 1	Cell Type 2	Cell Type 3	Cell Type 4	Cell Type 5	Cell Type 6
bandwidth 1.2	0.343	-0.003	0.280	0.844	0.838	0.931
bandwidth 2.2	0.329	-0.003	0.292	0.809	0.803	0.884
bandwidth 3.2	0.319	-0.003	0.285	0.742	0.736	0.801
bandwidth 4.2	0.314	-0.003	0.279	0.702	0.697	0.751

The color represents the Moran's I for all cell type, with higher correlation for redder and lower for greener.

Supplementary Table 8

2. SONAR depends on the initialization of pre-clustering. The authors need to show the robustness of initialization using other clustering methods. The reason is that Louvain is not standard in spatial transcriptomics data analysis.

Response: Thank you for your suggestions. Follow your suggestions, we evaluated four commonly used clustering algorithms, K-means, Louvain, Leiden and SLM (Smart local moving) on simulation datasets. We also compared different setting of parameters, i.e. number of clusters K (3, 4, 5, 6) for K-means and resolution parameter (0.1, 0.4, 0.7, 1) for the other three (**Supplementary Fig. 34**). The result shows that there are no significant differences between the results based on different pre-clustering algorithms. Such results demonstrate that SONAR is robust to the choice of pre-clustering algorithm.

We update the manuscript in **Methods** section in page 10 line 396:

“We have also evaluated several popular clustering methods with different settings of hyper-parameters, i.e. the resolution parameter. The results show that SONAR is robust to the choice of pre-cluster algorithms (Supplementary Note 5).”

In Supplementary Note 5:

“We performed a comparison of four clustering algorithms, K-means, Leiden, Louvain, SLM and applied different parameters ($k = 3, 4, 5, 6$ for K-means, resolution = 0.1, 0.4, 0.7, 1 for others) for each algorithm (Supplementary Fig. 34). We used the Kruskal-Wallis test to assess the differences among different conditions and found no significant difference. These results demonstrate the robustness of SONAR to the choice of pre-clustering algorithm and parameter. The robustness stems from the fact that pre-clustering only affects the selection of neighbor points, while the actual weights of each spot are determined by a combination of spatial kernel and elastic weighting, which makes the result more stable.”

Supplementary Fig. 34

Robust on different clustering algorithms and clustering parameters.

The boxplot shows the JSD for each spot under the K-means (k=3, 4, 5, 6, from left to right), Leiden, Louvain and SLM clustering algorithms with different resolution parameters (0.1, 0.4, 0.7, 1, from left to right) (n = 800)

3. In the simulations, the authors only considered using count matrix in real data, with limited information about simulation details. It would be more informative and helpful generating the simulated admixture of multiple cell types. Moreover, different ST technologies have distinct features in spatial resolution. It would be helpful to see the impact of resolution and spot scale in simulations. Other metrics such as Pearson's correlation should be evaluated at least in simulations.

Response: Thanks for your comment. We apologize for the unclear description of the simulation details.

To evaluate SONAR and compare it with other methods, we need to simulate spatial data with known cell type proportions at each spot. Therefore, we generate pseudo spatial data from an annotated peripheral blood mononuclear cell (PBMC) scRNA-seq dataset, and designed various cell type composition scenarios and spatial patterns. The expression mixture of multiple cell types at each simulated spatial spot is obtained with the following procedure: First, we simulate the observed number of cells for each cell type at the spot according to the designed composition (number) of cell types. We then randomly sample the corresponding number of cells of each type from the scRNA-seq dataset. Finally, we sum up the gene expression counts of all sampled cells to generate the mixed expression profile at that spot.

In the simulation based on real spatial datasets with single-cell resolution, such as mouse brain and human heart datasets, we divided the original coordinate space into grids. The expression profiles of all cells in each grid were summed up as the expression vector of each pseudo-spot, and the center point of the grid was used as the new coordinate. We have added these details in **Methods**

section (page 13 line 442):

“We generate pseudo spatial data from an annotated peripheral blood mononuclear cell (PBMC) scRNA-seq dataset, and designed various cell type composition scenarios and spatial patterns. The expression mixture of multiple cell types at each simulated spatial spot is obtained with the following procedure: First, we simulate the observed number of cells for each cell type at the spot according to the designed composition (number) of cell types. We then randomly sample the corresponding number of cells of each type from the scRNA-seq dataset. Finally, we sum up the gene expression counts of all sampled cells to generate the mixed expression profile at that spot (more details in Supplementary Note 6). ”

In page 14 line 487:

“Specifically, we divided the original coordinate space into grids. The expression profiles of all cells in each grid were summed up as the expression vector of each pseudo-spot, and the center point of the grid was used as the new coordinate. ”

Also, in light of your comment, we further investigate the impact of different resolution and spot scale using human heart spatial data (obtained by in situ sequencing), which containing 17,444 spots/cells. We originally gridded the data into pseudo-spots of size 454*424 pixels, which containing on average 16.5 cells. To assess the effect of spot scale, we further gridded the data with reduced sizes corresponding to 75% (356*318 pixels, with an average of 10 cells) and 50% (227*212 pixels, with an average of 4.9 cells) of the original grid size.

We evaluated the Pearson correlation (PCC) between the predicted proportion and the ground truth of each cell type over all pseudo-spots. SONAR maintained its advantage over the other compared methods, although there is a decrease in performance from the original grid size to the higher resolution size (**Supplementary Fig. 10a**). RCTD did not produce a result in the 50%-sized data due to an insufficient number of features (71.5 UMI counts per spot), but had comparable performance to SONAR on the other two datasets. In addition, we further evaluate the distribution of cNCCs and SPCs type, which were previously found to be predominantly distributed in the outflow tract region (OFT). In all resolutions tested, SONAR is the only method that can successfully recover the localization of these cell types (**Supplementary Fig. 10b,c**). SONAR also shows the highest PCCs between the estimated proportion of cNCCs and SPCs and the truth (**Supplementary Fig. 10b**).

We have updated these in **Methods** in page 15 line 496:

*“To assess the effect of spot scale, we further gridded the data with reduced sizes corresponding to 75% (356*318 pixels, with an average of 10 cells) and 50% (227*212 pixels, with an average of 4.9 cells) of the original grid size. ”*

In **Results** in page 6 line 213:

“ ” In addition, we further investigated the impact of different resolution and spot scale to the performance of deconvolution. We mimic pseudo-spots with higher resolution by reducing grid size to 75% and 50% of the original grid size (see Supplementary Note 2). SONAR maintained its

advantage over the other methods compared, although there is a decrease in performance from the original grid size to the higher resolution size (Supplementary Fig. 10a). In particular, the distributions of cNCCs and SPCs were robustly mapped by SONAR at different resolutions. (Supplementary Note 2 and Supplementary Fig. 10b,c).

In Supplementary Note 2:

*“We investigated the impact of different resolution and spot scale using human heart spatial data (obtained by in situ sequencing), which containing 17,444 spots/cells. We originally gridded the data into pseudo-spots of size 454*424 pixels, which containing on average 16.5 cells. To assess the effect of spot scale, we further gridded the data with reduced sizes corresponding to 75% (356*318 pixels, with an average of 10 cells) and 50% (227*212 pixels, with an average of 4.9 cells) of the original grid size.*

We evaluated the Pearson correlation (PCC) between the predicted proportion and the ground truth of each cell type over all pseudo-spots. SONAR maintained its advantage over the other compared methods, although there is a decrease in performance from the original grid size to the higher resolution size (Supplementary Fig. 10a). RCTD did not produce a result in the 50%-sized data due to an insufficient number of features (71.5 UMI counts per spot), but had comparable performance to SONAR on the other two datasets. In addition, we further evaluate the distribution of cNCCs and SPCs type, which were previously found to be predominantly distributed in the outflow tract region (OFT). In all resolutions tested, SONAR is the only method that can successfully recover the localization of these cell types (Supplementary Fig. 10b, c). SONAR also shows the highest PCCs between the estimated proportion of cNCCs and SPCs and the truth (Supplementary Fig. 10b). ”

Supplementary Fig. 10

SONAR show robust performance on different spot scale.

a, The boxplot shows the Pearson correlation between the predicted proportion and the ground truth on the whole region for all cell types under three different resolutions ($n = 12$). The color denotes datasets with different resolutions. **b**, The barplot shows the Pearson correlation between the predicted proportion and the ground truth on the whole region for Cardiac neural crest and Schwann progenitor cells under three different resolutions. **c**, The ground truth and scaled predicted proportions of Cardiac neural crest and Schwann progenitor cells inferred by comparing algorithms are displayed on each spot for three datasets with different resolution. Selected area is the magnified image of outflow tract(OFT). OGS is an abbreviation for original grid size. SD2 did not converge during the training of the graph model with datasets with reduced grid size. RCTD failed to return result in dataset with 50% of original grid size.

4. As a method paper, it would be more helpful using some widely used dataset in spatial transcriptomics, especially the ones in brain regions, which can be validated partially using information from Allen Brain Atlas. In addition, real data should include a larger dataset, e.g., the ones from slide-seq.

Response: We appreciate the reviewer's constructive suggestion. We additionally applied SONAR to a commonly used, larger and finer resolution dataset of mouse hippocampus, which was generated based on Slide-seqV2. We compared the predicted cell types with the corresponding

annotation in the Allen Brain Atlas and found that our method achieved high consistency (**Supplementary Fig. 8a**). Moreover, we computed the correlations of the predicted proportions and cell type markers for all cell types and found that our method outperformed other algorithms in overall cell type prediction (**Supplementary Fig. 8b, Supplementary Table 1 and 2**). We have revised the **Results** section in page 5 line 190:

“We further performed SONAR to mouse hippocampus dataset acquired by Slide-seqV2, which contains 41795 spots and 5093 genes. The mouse hippocampus has two major regions: Cornu Ammonis (CA) and Dentate Gyrus (DG). The principle cell type located by SONAR were in accordance to the regions annotated in Allen Brain Institute. For example, Dentate cell type was mainly located in the V-shaped granule layer in DG (Supplementary Fig. 8a). CA1 and CA3 cell types were correctly mapped to the corresponding pyramidal layers of CA. These pyramidal layers occupy a thin linear region, which is in the form of a typical background pattern (see Methods). Other methods such as RCTD, Cell2location and Stereoscope were also able to map these major cell types. When evaluating the correlation between inferred cell type proportions and cell type specific markers (Supplementary Table 1), SONAR showed superior performance over other methods in terms of overall correlation ranks. (Supplementary Fig. 8b and Supplementary Table 2).”

Data availability section in page 15 line 529:

The mouse hippocampus Slide-seqV2 dataset and annotated scRNA-seq data are available at https://singlecell.broadinstitute.org/single_cell/study/SCP948/robust-decomposition-of-cell-type-mixtures-in-spatial-transcriptomics.

Supplementary Fig. 8

SONAR accurately maps cell types to spatial location on mouse hippocampus Slide-seq V2 dataset.

a, Left: The annotation of hippocampus structures from the Allen Reference Atlas of an adult mouse brain. From bottom to top are CA1 sp, CA3 sp, Dentate Gyrus sg spatial domains in the Allen Reference Atlas. Right: The scaled predicted proportion of dominant cell types on each location inferred from SONAR and all comparing algorithms, including CA1, CA3, and Dentate cell types. **b**, The scatter plot shows the correlations between inferred cell-type proportions and corresponding cell-type-specific marker genes across spatial locations for all comparing algorithms. Rank (color) represents the type-specific descending order of Pearson correlation for all algorithms with the P value (size) tested by a single-sided (greater) test.

Supplementary Table 1: Markers list		
Cell types	genes	
Astrocyte	Slc1a3	Aqp4
CA1	Wfs1	
CA3	Hs3st4	
CR	Trp73	
Choroid	Folr1	
Dentate	Prox1	C1ql2
Endothelial Stalk	Cldn5	
Endothelial Tip	Nid1	
Ependymal	Ccdc153	
Microglia Macrophages	P2ry12	
Mural	Rgs5	Acta2
Neuron.Slc17a6	Slc17a6	
Oligodendrocyte	Mbp	
Polydendrocyte	Pdgfra	

Supplementary Table 1

Supplementary Table 2: Correlations between predicted proportion and cell type markers in SlideseqV2 dataset							
	SONAR	RCTD	CARD	SpatialDWLS	SPOTlight	Cell2location	Stereoscope
Astrocyte	0.230	0.248	0.212	0.155	0.233	0.196	0.241
CA1	0.291	0.171	0.186	0.336	0.111	0.217	0.216
CA3	0.165	0.162	0.221	0.241	0.199	0.251	0.145
Cajal Retzius	0.024	0.046	0.033	-0.003	0.031	0.028	0.014
Choroid	0.463	0.146	0.138	0.457	0.122	0.145	0.146
Denate	0.226	0.202	0.161	0.335	0.194	0.214	0.230
Endothelial Stalk	0.134	0.105	0.084	0.021	0.094	0.101	0.107
Endothelial Tip	0.111	0.058	0.042	0.089	0.058	0.051	0.035
Ependymal	0.434	0.225	0.209	0.261	0.183	0.211	0.226
Microglia Macrophages	0.258	0.188	0.161	0.161	0.166	0.173	0.190
Mural	0.193	0.221	0.069	0.051	0.137	0.179	0.216
Neuron.Slc17a6	0.072	0.150	0.157	0.011	0.091	0.201	-0.032
Oligodendrocyte	0.473	0.573	0.379	0.365	0.430	0.515	0.532
Polydendrocyte	0.180	0.139	0.097	0.086	0.125	0.113	0.123

The color represents the Spearman correlations by cell type in the table, with higher correlation for redder and lower for greener.

Supplementary Table 2

Other than those comments, I have some minor comments:

1. The order of subfigure a, b, c, d, e, f in figure 2 is not easy to understand. In general, it's order should be from left to right, or from top to bottom.

Response: Thank you for pointing out this issue. In the revised manuscript, we have modified Figure 2 and reordered the subfigures according to your advice.

2. Can you explain why you use the mean rank rather than only use RMSE or JSD, compared to the RMSE or JSD, what's the advantages of Mean rank over RMSE or JSD.

Response: We thank the reviewer for this comment. We use the mean rank as an aggregated metric to combine both RMSE and JSD. RMSE and JSD were used to measure different aspects of the model performance. RMSE (root mean square error) is a measure of the average distance between the predicted proportion and true proportion across all spots for each cell type (we have modified the RMSE according to reviewer2, see below for details). JSD (Jensen-Shannon divergence) is a measure of the similarity between the predicted cell type distribution and the true distribution at each spot. Therefore, we use the mean rank as a summary statistic that reflects both accuracy and similarity, and as a way to compare different models based on their relative performance. Since the values of RMSEs and JSDs may not be directly combinable, we believe that applying a rank-based aggregation provide a more robust metric for comparing model

performance.

We redefined RMSE according to reviewer2's comment. We now calculate RMSE for each cell type rather than for each spot in the previous version. Specifically, we calculated the distance between the predicted proportion and the true proportion of each cell type across all spots. The modified definition of RMSE, and accordingly the mean rank are revised in the manuscript as follows:

Methods section in page 12 line 425:

We define the RMSE of estimated cell type proportions as:

$$RMSE = \sqrt{\frac{1}{S} \sum_{s=1}^S (\tilde{u}_s^t - u_s^t)^2}$$

Where S is the total number of all spots, \tilde{u}_s^t is the predicted proportion for type t in spot s , u_s^t is the ground truth of cell type t . The lower RMSE means the better prediction.

Methods section in page 13 line 433:

$$Rank_{RMSE} = \frac{1}{T} \sum_{t \in \tau} Rank_{RMSE}^t$$
$$Rank_{JSD} = \frac{1}{S} \sum_{s \in D} Rank_{JSD}^s$$

where T represents the total number of cell types, S represents the total number of spots, $Rank_{RMSE}^t$ and $Rank_{JSD}^s$ are the ranks of RMSE and JSD for the algorithm on cell type t and spot s , respectively.

3, In github, can you explain how to use your software in detail. For example give a brief example to show how to use your software like

BayesSpace <https://edward130603.github.io/BayesSpace/articles/BayesSpace.html>

Response: We appreciate your advice! We apologize for missing explicit examples in the github.

We have now updated the github page and added an example is available at

<https://github.com/lzygenomics/SONAR/blob/master/Example/SONAR-entrance.Rmd>.

4. The word "composition" should be composition in Figure 5e caption

Response: We appreciate your careful attention to the detail. We have corrected this typo in the caption of Figure 5e.

5. In this paper, the author introduces methods SONAR with two versions (SONAR and SONAR-0). In the real data analysis, how do we choose them, is there an automatic way to choose it.

Response: We thank the reviewer for this question. The default version on our GitHub page is SONAR. SONAR-0 is just a simplified version of SONAR that does not use pre-clustering and elastic weighting. It serves as a baseline model for applying the GWR framework for comparison. We have shown, by simulation and real data, that SONAR outperforms SONAR-0 in almost all scenarios and is more robust. Therefore, we will always recommend using SONAR in real data

analysis.

6. Some terms in the reference need some change. For example, in the reference 6, slide-seqv2 should be Slide-seqV2. In the reference 13, dna should be DNA. In the reference 14, 19, rna should be RNA. In the reference 44, rna-seq should be RNA-Seq. In the reference 52,53, t cell should be T cell. In the reference 50, b cells should be B cells, cd40/cd154 should be CD40/CD154.

Response: Thank you for pointing out our mistakes. We have corrected these typos in the **References** section of the manuscript.

Reviewer #2 (Remarks to the Author):

This study introduced a computational method called SONAR for cellular deconvolution problem of spatial transcriptomics data by using a spatially weighted Poisson-gamma regression model. SONAR utilized an elastic weighting step to adaptively filter dissimilar neighbors which could identify the boundary of different regions more accurately. SONAR has been benchmarked on different datasets and the performance was over the other state-of-the-art methods. Overall, this study showed novelty in method development and conducted comprehensive experiments on different datasets. However, there are still several issues that need to be addressed.

Response: We are very pleased that the reviewer considers our study showed novelty in method development and conducted comprehensive experiments on different datasets. We greatly appreciated the constructive suggestions from the reviewer. We have addressed all comments with the point-by-point responses as follows.

1. The authors claimed that CARD was the only deconvolution method that utilized spatial information of spots. But to my knowledge, there are also some methods used the spatial information including SpiceMix, SD2, SpaOTsc and novoSpaRc. The authors should also discuss about and compare with these studies.

Response: We thank the reviewer for the suggestion. We agree that SpiceMix, SD2, SpaOTsc and novoSpaRc all use spatial information in their models. However, we did not compare SpaOTsc, novoSpaRc and SpiceMix in our study because they are not typically designed for cell type deconvolution task. SpiceMix is a latent variable model that try to elucidate the interplay of spatial and intrinsic factors of each spot. It is an unsupervised method that learns the set of latent factors (metagenes) and represent each spot by a mixture of metagenes. SpaOTsc and novoSpaRc are optimal-transport based modeling that use single-cell RNAseq data. novoSpaRc does not even rely on gene expression profiles in the spatial spots. It reconstructs spatial gene expression from single-cell data based on the structural correspondence hypothesis that cells in physical proximity share similar gene expression profiles. SpaOTsc is specifically designed for image-based spatial expression data. It aims to complete the expression profile of each cell/spot in space and infer cell-cell communications.

SD2 is a graph convolutional neural network (GCN) based method that incorporates dropout and spatial information for deconvolution. We have added the comparisons with SD2 in our revised manuscript and updated the corresponding results in **Fig. 2b, c, d** and **Fig. 3b, c** and **Fig. 4c, e** and **Supplementary Figure 2, 4, 5, 6, 9, 10, 11, 12, 13, 14, 32** and **Supplementary Table 4, 5**. The results show that SONAR consistently outperforms SD2 on both simulation and real datasets. Moreover, SD2 failed to run on human heart data with reduced grid size. In addition, it did not converge in training the graph model in mouse hippocampus and liver cancer datasets. We have added these details into the revised manuscript.

We also added a paragraph in **Discussion** section in Page 8, Line 322:

“There are methods that make use of spatial information and aim to perform spatial expression reconstruction based on scRNA-seq data. For example, the optimal-transport based models novoSpaRc and SpaOTsc. novoSpaRc reconstructs spatial gene expression from single-cell data to a predefined physical region, based on a structural correspondence hypothesis, that is

cells in physical proximity share similar gene expression profiles. SpaOTsc is specifically designed for image-based spatial expression data. It aims to complete the expression profile of each cell/spot in space and infer cell-cell communications. Other deconvolution algorithms such as SpiceMix, unsupervisedly represents each spot by a mixture of latent factors (metagenes) and try to elucidate the interplay of spatial and intrinsic factors of each spot. Although these methods map cell expression profile to spatial locations or deconvolute spatial transcriptome into latent factors, they are not typically designed for spatial cell type deconvolution task. As a result, we did not compare with these methods.”

2. In the experiments of Developing human heart data, it is great to see that SONAR could identify tiny structures like cNCCs and SPCs. I wonder if there would be the similar results when authors grid the cells into different resolutions of spots. Authors could try different resolutions of spots and use the quantified metrics to evaluate the performance which would directly show the benchmarking results through different methods.

Response: Thank you for the suggestion. We originally gridded the Developing human heart data into pseudo-spots of size 454*424 pixels, which containing on average 16.5 cells. In the revised manuscript, we further gridded the data with reduced sizes corresponding to 75% (356*318 pixels, with an average of 10 cells) and 50% (227*212 pixels, with an average of 4.9 cells) of the original grid size.

We evaluated the Pearson correlation (PCC) between the predicted proportion and the ground truth of each cell type over all pseudo-spots. SONAR maintained its advantage over the other compared methods, although there is a decrease in performance from the original grid size to the higher resolution size (**Supplementary Fig. 10a, b**). In addition, we further evaluate the distribution of cNCCs and SPCs type, which were previously found to be predominantly distributed in the outflow tract region (OFT). In all resolutions tested, SONAR maintains the only method that can successfully recover the localization of these cell types. SONAR also shows the highest PCCs between the estimated proportion of cNCCs and SPCs and the truth (**Supplementary Fig. 10b, c**). The results demonstrated the ability of SONAR to capture the fine spatial structure of cell composition.

We have updated **Methods** section in page 15 line 496:

*“To assess the effect of spot scale, we further gridded the data with reduced sizes corresponding to 75% (356*318 pixels, with an average of 10 cells) and 50% (227*212 pixels, with an average of 4.9 cells) of the original grid size. ”*

In **Results** section in page 6 line 213:

“ ” In addition, we further investigated the impact of different resolution and spot scale to the performance of deconvolution. We mimic pseudo-spots with higher resolution by reducing grid size to 75% and 50% of the original grid size (see Supplementary Note 2). SONAR maintained its advantage over the other methods compared, although there is a decrease in performance from the original grid size to the higher resolution size (Supplementary Fig. 10a). In particular, the distributions of cNCCs and SPCs were robustly mapped by SONAR at different resolutions.

(Supplementary Note 2 and Supplementary Fig. 10b,c).

In Supplementary Note 2:

“We investigated the impact of different resolution and spot scale using human heart spatial data (obtained by in situ sequencing), which containing 17,444 spots/cells. We originally gridded the data into pseudo-spots of size 454*424 pixels, which containing on average 16.5 cells. To assess the effect of spot scale, we further gridded the data with reduced sizes corresponding to 75% (356*318 pixels, with an average of 10 cells) and 50% (227*212 pixels, with an average of 4.9 cells) of the original grid size.

We evaluated the Pearson correlation (PCC) between the predicted proportion and the ground truth of each cell type over all pseudo-spots. SONAR maintained its advantage over the other compared methods, although there is a decrease in performance from the original grid size to the higher resolution size (Supplementary Fig. 10a). RCTD did not produce a result in the 50%-sized data due to an insufficient number of features (71.5 UMI counts per spot), but had comparable performance to SONAR on the other two datasets. In addition, we further evaluate the distribution of cNCCs and SPCs type, which were previously found to be predominantly distributed in the outflow tract region (OFT). In all resolutions tested, SONAR is the only method that can successfully recover the localization of these cell types (Supplementary Fig. 10b, c). SONAR also shows the highest PCCs between the estimated proportion of cNCCs and SPCs and the truth (Supplementary Fig. 10b).”

Supplementary Fig. 10

SONAR show robust performance on different spot scale.

a, The boxplot shows the Pearson correlation between the predicted proportion and the ground truth on the whole region for all cell types under three different resolutions ($n = 12$). The color denotes datasets with different resolutions. **b**, The barplot shows the Pearson correlation between the predicted proportion and the ground truth on the whole region for Cardiac neural crest and Schwann progenitor cells under three different resolutions. **c**, The ground truth and scaled predicted proportions of Cardiac neural crest and Schwann progenitor cells inferred by comparing algorithms are displayed on each spot for three datasets with different resolution. Selected area is the magnified image of outflow tract(OFT). OGS is an abbreviation for original grid size. SD2 did not converge during the training of the graph model with datasets with reduced grid size. RCTD failed to return result in dataset with 50% of original grid size.

3. Since authors simulated the pseudo-spots by randomly sampling the cells from PBMC scRNA-seq datasets, how was the Tobler's first law of geography authors mentioned embodied in the simulated dataset? The neighbored pseudo-spots should have similar expression profiles. Authors should consider this in a more systematic way and regulate the pipeline of simulation.

Response: We appreciate the reviewer's thoughtful question. The Tobler's first law of geography states that, "everything is related to everything else, but near things are more related than distant things.". In the simulations of Compo-Areas (layer or block patterns), we aim to imitate patchy landscapes. Spots within the same patch represent "near things" and their composition of cell types are likely to be more related. Spots across different patches represent "distant things" which have cell compositions sampled from different distribution. These designs imply, to some extent, a discrete form of Tobler's law.

In addition, we also simulated (Compo Areas) scenarios with Gradient transition mode between regions (patches), which resemble the diffusion of cell types. Such setting is used to represent the Tobler's law in fine-scale (continuous). We show in the exemplar data in **Fig. R1** that the (Euclidean) distance of cell type composition between spots increase as physical distance increase (along the axis with gradient change). The tendency of increasing compositional distance with physical distance was also observed in real (Mouse brain) ST data (**Fig. R2**). Moreover, SONAR had overall better performance in simulated data with such gradient transition mode (**Fig. R1b**).

We hope these revisions can address the reviewer's concern. Thanks for reviewer's insightful question again!

Fig. R1

a. Changes in similarity with increasing distance in Gradient simulations

We expanded the set of pseudo-spots by distance with a pseudo-spot column as reference, and measured the similarity between them by calculating the Euclidean distance of the type composition. **In left:** The graph showed pseudo-spots in the transition part of Gradient

simulations. The selected column was as reference, b1 was the spots set including left 2 columns, b2 was the spots set including left 3 columns, ..., b9 was the spots set including left 9 columns. **In right:** the boxplot showed the Euclidean distance of the cell type composition for each pair of spots between the b_i ($i = 1, \dots, 9$) spot set and references column. The physical distance to reference column increased from b1 to b9.

b. Performance on transition part in Gradient simulations

The boxplot showed RMSE by each cell type and JSD by each spot on transition part in the Gradient simulations, including 5 replicates. Each box on the top contains 40 (8 cell types*5 replicates) points, each box on the bottom contains 4000 (800 spots*5 replicates) points.

Fig. R2

Sharp changes in similarity with increasing distance in mouse brain

In Left: The scatter points represent the true proportions of Excitatory L6 cells in each gridded spot of the mouse brain spatial transcriptomics data. The reference column is the selected column. The other 10 columns from the right side of reference are labeled b1 to b10, where b1 is adjacent to the reference column and b10 is the farthest to the right. **In right:** The boxplot showed the Euclidean distance of the cell type composition for each pair of spots between the b_i ($i = 1, \dots, 10$) spot set and references column. The physical distance to reference column increased from b1 to b10.

4. In the benchmarking experiments, authors could record the computational time to test the efficiency of SONAR and other compared methods.

Response: We appreciate the reviewer’s suggestion. We have added the computational time of SONAR and other compared methods on a simulation dataset, a Visium dataset and a large-scale SlideseqV2 dataset. We have added these results in **Discussion** section and **Supplementary Note 3** as follows:

In **Discussion** section in page 8 line 320:

“In addition, SONAR demonstrated high computational efficiency (Supplementary Note 3).”

Supplementary Note 3:

“To show the computation efficiency of SONAR, we recorded the computational time of SONAR and other compared methods on a simulation datasets, a Visium dataset (human liver HCC) and a large-scale SlideseqV2 dataset (mouse hippocampus). The simulation dataset consists of 800 spots and 2845 genes, the Visium dataset has 2,791 spots and 17,735 genes and the SlideseqV2 dataset has 41795 spots and 5093 genes. Computation of SONAR, RCTD, CARD, SPOTlight and SpatialDWLS were on the CPU processor: Intel(R) Xeon(R) Gold 6248R CPU @ 3.00GHz processor. Computation of Cell2location, Stereoscope and SD2 were performed on a RTX 3090 GPU processor. The results show that SONAR has a high computational efficiency in all tested datasets (Supplementary Fig. 32).”

Supplementary Fig. 32

Computational time of comparing algorithms

a, the bar plot shows the computational time for the simulation datasets which contains 800 spots and 2,845 genes. **b**, the computational time for a liver cancer(HCC) Visium dataset, which contains 2,791 spots and 17,735 genes. **c**, the computational time for a large-scale SlideseqV2 dataset, which contains 41,795 spots and 5,093 genes. The computation time was in units of seconds in **a**, and in units of minutes **b** and **c**.

5. The authors said ‘We also discovered a significantly lower proportion of Myeloid cells in the Normal region compared to the Tumor region only in sample HCC-2L (Supplementary Fig. 17). These regional distinctions between and within samples were in accordance with (25), but were not fully characterized by other methods (Supplementary Fig. 19-24).’ But as the figures showed, CARD, RCTD and stereoscope also had the similar results of Myeloid cells in HCC-2L. Authors should explain this inconsistency between the statements and figures.

Response: We thank the reviewer for pointing out this issue. We apologize for the confusion caused by our misdescription. When we stated that “We also discovered a significantly lower proportion of Myeloid cells in the Normal region compared to the Tumor region only in sample HCC-2L (Supplementary Fig. 17).”, we intended to emphasize that: SONAR revealed a lower proportion of Myeloid cells in the normal region than in the tumor region only in HCC-2L,

whereas it revealed a higher proportion of Myeloid cells in the normal region than in the tumor region in HCC-1L, 3L and 4L. Although CARD and Stereoscope showed a similar result for HCC-2L, they did not show a consistent result across the other samples. Specifically, CARD missed the trend in HCC-4L, and stereoscope missed the trend in HCC-1L and 4L (Fig. R3). Moreover, we acknowledge our misdescription for RCTD. We have revised this part in the manuscript:

In **Results** section in page 7 line 273:

“We also discovered a significantly lower proportion of Myeloid cells in the Normal region compared to the Tumor region only in sample HCC-2L (Supplementary Fig. 17), while the opposite trend was observed in the other samples. These regional distinctions between and within samples were consistent with (25), and were fully characterized only by SONAR and RCTD (Supplementary Fig. 19-24 Note that we excluded SD2 from this comparison since its training process did not converge in all four datasets).”

Fig. R3 (for convenient, we selected corresponding parts in Supplementary Fig.19-24)

Predicted proportion of Myeloid in 4 patients for Normal and Tumor region

The boxplot shows the predicted proportions of Myeloid cells in the normal and tumor regions of four patients by different algorithms. The evidence showed that the significantly lower proportion

of Myeloid cells in the normal region than in the tumor region only in HCC-2L, whereas the opposite trend was observed in HCC-1L, 3L and 4L. Therefore, we performed a one-sided Wilcoxon test with the alternative hypothesis of “greater” in HCC-1L, 3L and 4L, and “less” in HCC-2L. ****: $p < 0.0001$, *: $p < 0.05$, ns: $p > 0.05$.

6. In the experiments of primary liver cancer, authors could visualize the results of SONAR and other compared methods which will give the intuitive results to prove that SONAR could fine-mapping the sharp boundary regions in heterogeneous tissues.

Response: We thank you for your constructive suggestions. In light of your comment, we mapped the regional median LoCo score (Fibroblast and B cells) onto the spatial spots. It allowed us to explicitly visualize the sharp outer edges resolved by SONAR that form a local peak over neighboring regions. We have updated these results in the revised manuscript, Supplementary Figs.28-30 and Supplementary Table 7.

In **Results** section in page 8 line 302:

“In particular, mapping the regional median LoCo scores (Supplementary Fig. 28-30 and Supplementary Table 7) onto the spatial spots, allowed us to explicitly visualize the sharp outer edges resolved by SONAR that form a local peak over neighboring regions”

Supplementary Fig. 28

Median LoCo Score of HCC-2L for all Spot Types

Supplementary Fig. 29

Supplementary Fig. 30

Median LoCo scores of Fibroblasts and B cells by Spot Type for in HCC-1L/2L/4L.

The top, left figure show the Spot Type annotation (same as Figure 5e), different colors represent different Spot Type. Specifically, the spots are stratified according to the regional composition of their neighboring set into 7 types: Pure Normal, Outer edge to Transition region (Normal or Tumor side), Inner edge to Transition region (Normal or Tumor side), Interior Transition region, and Pure Tumor region. Other figures show the median (by Spot Type) LoCo scores of Fibroblasts and B cells on the spatial spots for SONAR, Cell2location, SPOTlight, RCTD, Stereoscope, CARD and SpatialDWLS. The color bar represents the value of LoCo scores.

Supplementary Table 7: Median of LoCo scores of Fibroblasts and B Cells by Spot Type									
Method	Patient	Trend	Median of LoCo scores by Spot Type						
			Pure Normal	Outer Edge (N.2Tr.)	Inner Edge (Tr.2N.)	Interior	Inner Edge (Tr.2Tu.)	Outer Edge (Tu.2Tr.)	Pure Tumor
SONAR	HCC-1L		0.383	0.738	0.549	0.069	0.386	0.608	0.067
	HCC-2L		0.183	0.217	0.000	-0.133	0.321	0.533	0.333
	HCC-4L		0.200	0.467	0.275	-0.067	0.367	0.400	0.150
Cell2location	HCC-1L		0.267	0.577	0.548	0.150	0.617	0.646	0.167
	HCC-2L		0.133	0.200	0.267	0.257	0.500	0.571	0.333
	HCC-4L		0.067	0.217	0.117	-0.017	0.192	0.405	0.190
SPOTlight	HCC-1L		0.217	0.576	0.533	0.167	0.377	0.208	-0.050
	HCC-2L		0.467	0.417	0.233	0.095	0.217	0.233	0.133
	HCC-4L		0.400	0.483	0.333	0.400	0.362	0.167	0.033
RCTD	HCC-1L		0.333	0.700	0.550	0.067	0.433	0.600	0.050
	HCC-2L		0.187	0.267	0.033	-0.017	0.429	0.467	0.267
	HCC-4L		0.179	0.400	0.217	-0.050	0.400	0.357	0.104
Stereoscope	HCC-1L		0.000	0.467	0.350	-0.396	-0.258	0.086	-0.100
	HCC-2L		0.000	-0.033	-0.267	-0.400	-0.100	0.200	0.000
	HCC-4L		-0.143	0.117	-0.033	-0.429	-0.083	-0.100	-0.133
CARD	HCC-1L		0.857	0.883	0.717	0.242	0.695	0.950	0.933
	HCC-2L		0.683	0.483	0.150	-0.017	0.300	0.333	0.233
	HCC-4L		0.867	0.883	0.533	-0.071	-0.067	0.367	0.500
SpatialDWLS	HCC-1L		NA	0.561	0.240	-0.274	-0.046	NA	NA
	HCC-2L		-0.046	0.040	-0.075	-0.183	0.157	0.325	0.014
	HCC-4L		1.000	0.959	0.259	-0.109	0.000	0.540	NA

Supplementary Table 7

7. Have authors tried large-scale spatial transcriptomics datasets, such as Slide-seq V2 which may contain more than tens of thousands of spots? I wonder about the performance of SONAR on large-scale datasets compared with other state-of-the-art methods.

Response: We appreciate your constructive suggestion. Follow your suggestion, we applied our method and other algorithms to a commonly used, larger and finer resolution dataset of mouse hippocampus based on Slide-seqV2. The cell types predicted by SONAR were correctly mapped to the corresponding subregions in the mouse hippocampus according to the annotation in the Allen Brain Atlas (**Supplementary Fig. 8a**). When evaluating the correlation between predicted cell type proportions and cell type specific markers, SONAR was in the top tier of all compared methods (**Supplementary Fig. 8b**). We have added these results into the revision accordingly:

We have revised the **Results** section in page 5 line 190:

“We further performed SONAR to mouse hippocampus dataset acquired by Slide-seqV2, which contains 41795 spots and 5093 genes. The mouse hippocampus has two major regions: Cornu Ammonis (CA) and Dentate Gyrus (DG). The principle cell type located by SONAR were in accordance to the regions annotated in Allen Brain Institute. For example, Dentate cell type was mainly located in the V-shaped granule layer in DG (Supplementary Fig. 8a). CA1 and CA3 cell types were correctly mapped to the corresponding pyramidal layers of CA. These pyramidal layers occupy a thin linear region, which is in the form of a typical background pattern (see Methods). Other methods such as RCTD, Cell2location and Stereoscope were also able to map these major cell types. When evaluating the correlation between inferred cell type proportions and cell type specific markers (Supplementary Table 1), SONAR showed superior performance over other methods in terms of overall correlation ranks. (Supplementary Fig. 8b and Supplementary Table 2).”

Data availability section in page 15 line 529:

The mouse hippocampus Slide-seqV2 dataset and annotated scRNA-seq data are available at https://singlecell.broadinstitute.org/single_cell/study/SCP948/robust-decomposition-of-cell-type-mixtures-in-spatial-transcriptomics.

Supplementary Fig. 8

SONAR accurately maps cell types to spatial location on mouse hippocampus Slide-seq V2 dataset.

a, Left: The annotation of hippocampus structures from the Allen Reference Atlas of an adult mouse brain. From bottom to top are CA1 sp, CA3 sp, Dentate Gyrus sg spatial domains in the Allen Reference Atlas. Right: The scaled predicted proportion of dominant cell types on each location inferred from SONAR and all comparing algorithms, including CA1, CA3, and Dentate cell types. **b**, The scatter plot shows the correlations between inferred cell-type proportions and corresponding cell-type-specific marker genes across spatial locations for all comparing algorithms. Rank (color) represents the type-specific descending order of Pearson correlation for all algorithms with the P value (size) tested by a single-sided (greater) test.

Supplementary Table 1: Markers list		
Cell types	genes	
Astrocyte	Slc1a3	Aqp4
CA1	Wfs1	
CA3	Hs3st4	
CR	Trp73	
Choroid	Folr1	
Dentate	Prox1	C1ql2
Endothelial Stalk	Cldn5	
Endothelial Tip	Nid1	
Ependymal	Ccdc153	
Microglia Macrophages	P2ry12	
Mural	Rgs5	Acta2
Neuron.Slc17a6	Slc17a6	
Oligodendrocyte	Mbp	
Polydendrocyte	Pdgfra	

Supplementary Table 1

Supplementary Table 2: Correlations between predicted proportion and cell type markers in SlideseqV2 dataset							
	SONAR	RCTD	CARD	SpatialDWLS	SPOTlight	Cell2location	Stereoscope
Astrocyte	0.230	0.248	0.212	0.155	0.233	0.196	0.241
CA1	0.291	0.171	0.186	0.336	0.111	0.217	0.216
CA3	0.165	0.162	0.221	0.241	0.199	0.251	0.145
Cajal Retzius	0.024	0.046	0.033	-0.003	0.031	0.028	0.014
Choroid	0.463	0.146	0.138	0.457	0.122	0.145	0.146
Denate	0.226	0.202	0.161	0.335	0.194	0.214	0.230
Endothelial Stalk	0.134	0.105	0.084	0.021	0.094	0.101	0.107
Endothelial Tip	0.111	0.058	0.042	0.089	0.058	0.051	0.035
Ependymal	0.434	0.225	0.209	0.261	0.183	0.211	0.226
Microglia Macrophages	0.258	0.188	0.161	0.161	0.166	0.173	0.190
Mural	0.193	0.221	0.069	0.051	0.137	0.179	0.216
Neuron.Slc17a6	0.072	0.150	0.157	0.011	0.091	0.201	-0.032
Oligodendrocyte	0.473	0.573	0.379	0.365	0.430	0.515	0.532
Polydendrocyte	0.180	0.139	0.097	0.086	0.125	0.113	0.123

The color represents the Spearman correlations by cell type in the table, with higher correlation for redder and lower for greener.

Supplementary Table 2

8. Following current definition of RMSE, some cell types with high abundance may dominate the metric which means it may favor an algorithm that predicts well cell types with high proportion. Authors need to redefine the RMSE.

Response: We appreciate your valuable comment. In the previous version, we use RMSE (root mean square error) to measure average distance between the predicted cell type proportion and the actual proportion across all types at a spot. The metric may be dominated by high abundance cell, which means it may favor algorithms that predicts well cell types with high proportion. Following your suggestion, we redefined the RMSE. We now calculate RMSE for each cell type rather than for each spot. Specifically, we calculated the average distance between the predicted proportion and the true proportion of each cell type across all spots. Thus, each dataset will now have T RMSE values (T = number of cell types).

In the revised manuscript, we have modified the definition of RMSE and RMSE based Mean rank as follows:

Methods section in page 12 line 425:

We define the RMSE of estimated cell type proportions as:

$$\text{RMSE} = \sqrt{\frac{1}{S} \sum_{s=1}^S (\tilde{u}_s^t - u_s^t)^2}$$

Where S is the total number of all spots, \tilde{u}_s^t is the predicted proportion for type t in spot s , u_s^t is the ground truth of cell type t . The lower RMSE means the better prediction.

Methods section in page 13 line 433:

$$\text{Rank}_{\text{RMSE}} = \frac{1}{T} \sum_{t \in \tau} \text{Rank}_{\text{RMSE}}^t$$

$$\text{Rank}_{\text{JSD}} = \frac{1}{S} \sum_{s \in D} \text{Rank}_{\text{JSD}}^s$$

where T represents the total number of cell types, S represents the total number of spots, $\text{Rank}_{\text{RMSE}}^t$ and $\text{Rank}_{\text{JSD}}^s$ are the ranks of RMSE and JSD for the algorithm on cell type t and spot s , respectively.

We also updated the corresponding results (**Fig. 2c, d; Fig. 3b, c; Supplementary Fig. 2a, b, d, f; Supplementary Fig. 4b, d, f, h, j**).

9. In the pipeline of pre-clustering, authors claimed that they used the Louvain algorithm. I wonder which resolution authors used. Because the choice of resolution affects the number of clusters. Is it fixed in the pipeline or regulated manually as the hyper parameter through each experiment?

Response: Thank you for your suggestions. Following your suggestion, we evaluated four commonly used clustering algorithms, K-means, Louvain, Leiden and SLM (Smart local moving) on simulation datasets. We also compared different setting of parameters, i.e. number of clusters K (3, 4, 5, 6) for K-means and resolution parameter (0.1, 0.4, 0.7, 1) for the other three (**Supplementary Fig. 34**). The result shows that there are no significant differences between the results based on different pre-clustering algorithms. Such results demonstrate that SONAR is robust to the choice of pre-clustering algorithm.

We update the manuscript in **Methods** section (page 10 line 396):

“We have also evaluated several popular clustering methods with different settings of hyper-parameters, i.e. the resolution parameter. The results show that SONAR is robust to the choice of pre-cluster algorithms (Supplementary Note 5).”

In Supplementary Note 5:

“We performed a comparison of four clustering algorithms, K-means, Leiden, Louvain, SLM and applied different parameters ($k = 3, 4, 5, 6$ for K-means, resolution = 0.1, 0.4, 0.7, 1 for others) for each algorithm (Supplementary Fig. 34). We used the Kruskal-Wallis test to assess the differences among different conditions and found no significant difference. These results demonstrate the robustness of SONAR to the choice of pre-clustering algorithm and parameter. The robustness stems from the fact that pre-clustering only affects the selection of neighbor points, while the actual weights of each spot are determined by a combination of spatial kernel and

elastic weighting, which makes the result more stable.”

Supplementary Fig. 34

Robust on different clustering algorithms and clustering parameters.

The boxplot shows the JSD for each spot under the K-means (k=3, 4, 5, 6, from left to right), Leiden, Louvain and SLM clustering algorithms with different resolution parameters (0.1, 0.4, 0.7, 1, from left to right) (n = 800)

REVIEWERS' COMMENTS

Reviewer #1 (Remarks to the Author):

The authors have thoroughly addressed my comments. In particular, they have performed additional benchmarks on both simulation and existing datasets, including applying SONAR to Slide-seqV2 data of mouse hippocampus and comparing it to the Allen Brain Atlas, evaluating the impact of pre-clustering methods and the resolution of spatial spots, and assessing the computational efficiency. They also addressed my minor comment regarding the metrics they used for evaluation. Finally, they have adjusted the display of Figure 2 and updated their github page with a brief example. The authors have addressed all my previous concerns and have made great efforts to revise the manuscript.

I only have very few minor comments:

1. The word “celluar” should be “cellular” in the introduction lines 30, 31 and 70.
2. In line 321, “Supplementary” should be “Supplementary”.
3. In reference 8, an http link should at least be included and cited.
4. The method names in references 15,19, 20 and 23, should be “SpatialDWLS”, “SCDC”, “DecOT”, “stLearn”, “BayesSpace”, instead of “SpatialdwlS”, “Scdc”, “Decot”, “stlearn”, “bayesspace”.
5. In reference 59, “poisson” should be “Poisson”

Reviewer #2 (Remarks to the Author):

The authors have addressed my comments.

We are very grateful to the reviewers for their thorough comments. We have revised the manuscript following all the comments. Below please find the point-by-point response to all the reviewers' comments. In this response letter: the black text is the reviewer's comment, the blue text is my reply to the above comment.

Point-by point Responses to Reviewers' Comments:

Reviewer #1 (Remarks to the Author):

The authors have thoroughly addressed my comments. In particular, they have performed additional benchmarks on both simulation and existing datasets, including applying SONAR to Slide-seqV2 data of mouse hippocampus and comparing it to the Allen Brain Atlas, evaluating the impact of pre-clustering methods and the resolution of spatial spots, and assessing the computational efficiency. They also addressed my minor comment regarding the metrics they used for evaluation. Finally, they have adjusted the display of Figure 2 and updated their github page with a brief example. The authors have addressed all my previous concerns and have made great efforts to revise the manuscript.

I only have very few minor comments:

1. The word "celluar" should be "cellular" in the introduction lines 30, 31 and 70.
2. In line 321, "Supplementary" should be "Supplementary".
3. In reference 8, an http link should at least be included and cited.
4. The method names in references 15,19, 20 and 23, should be "SpatialDWLS", "SCDC", "DecOT", "stLearn", "BayesSpace", instead of "Spatialdwls", "Scdc", "Decot", "stlearn", "bayesspace".
5. In reference 59, "poisson" should be "Poisson"

Response: We thank the reviewer for his positive evaluation and meticulous examination of our work. Based on comments 1 and 2, we have corrected typos "cellular" and "Supplementary", and based on comments 3,4,5 as well as checklist, we have revised the Reference section and corrected the titles.

Reviewer #2 (Remarks to the Author):

The authors have addressed my comments.

Response: We would again like to express our appreciation for the reviewer's time and effort. The manuscript has strengthened considerably as a result of their efforts